# DECOUPLED CLASSIFIER-FREE GUIDANCE FOR COUNTERFACTUAL DIFFUSION MODELS

## ABSTRACT

Counterfactual generation aims to simulate realistic hypothetical outcomes under causal interventions. Diffusion models have emerged as a powerful tool for this task, combining DDIM inversion with conditional generation and classifier-free guidance (CFG). In this work, we identify a key limitation of CFG for counterfactual generation: it prescribes a global guidance scale for all attributes, leading to significant spurious changes in inferred counterfactuals. To mitigate this, we propose *Decoupled Classifier-Free Guidance* (DCFG), a flexible and model-agnostic guidance technique that enables attribute-wise control following a causal graph. DCFG is implemented via a simple attribute-split embedding strategy that disentangles semantic inputs, enabling selective guidance on user-defined attribute groups. Our experiments demonstrate that DCFG significantly improves the axiomatic soundness of inferred counterfactuals on challenging medical imaging data, mitigating spurious amplification effects, and enhancing counterfactual reversibility.

## 1 INTRODUCTION

Counterfactual generation is considered to be fundamental to causal reasoning (Pearl, 2009; Peters et al., 2017; Bareinboim et al., 2022), allowing us to explore hypothetical scenarios such as: *'How would this patient's disease have progressed if they had received treatment A instead of treatment B?'*. Answering such causal questions is important across various domains, such as healthcare (Castro et al., 2020), fairness (Kusner et al., 2017) and scientific discovery (Narayanaswamy et al., 2020). There has been a growing interest in generating counterfactual images using deep generative models, aiming to simulate how visual data would change under hypothetical interventions. Recent works build Structural Causal Models (SCMs) (Pearl, 2009) using deep generative model components such as normalizing flows (Rezende & Mohamed, 2015), Variational Autoencoders (VAEs) (Kingma & Welling, 2013; Child, 2020) and diffusion models (Sohl-Dickstein et al., 2015a; Ho et al., 2020; Ribeiro et al., 2025), enabling principled counterfactual inferences via *abduction-action-prediction* (Pawlowski et al., 2020; Sanchez & Tsaftaris, 2022; Ribeiro et al., 2023; Wu et al., 2025; Rasal et al., 2025).

Diffusion models have emerged as the state-of-the-art approach for image synthesis, achieving unprecedented fidelity and perceptual quality (Dhariwal & Nichol, 2021; Podell et al., 2023). Many previous works have explored diffusion models for counterfactual generation (Sanchez et al., 2022b;a; Pérez-García et al., 2024; Komanduri et al., 2024; Rasal et al., 2025; Kumar et al., 2025), leveraging Denoising Diffusion Implicit Models (DDIM) (Song et al., 2020) to deterministically encode images into a latent space, followed by conditional generation with modified attributes. Conditioning is typically enforced through discriminative score functions, either with external classifiers (Dhariwal & Nichol, 2021) or through Classifier-free Guidance (CFG) (Ho & Salimans, 2022). Combining DDIM inversion and guided conditional decoding has also become the dominant paradigm in diffusion-based image editing (Couairon et al., 2022; Wallace et al., 2023; Hertz et al., 2022; Epstein et al., 2023).

In counterfactual generation, CFG plays a crucial role in ensuring that interventions are *effective*, i.e. that the intended changes are reflected in the output. While recent works have proposed refinements to CFG to enhance fidelity (Chung et al., 2024; Kynkäänniemi et al., 2024), we identify that CFG exacerbates spurious effects of image attributes that should remain stable under causal interventions, a phenomenon known as *attribute amplification* (Xia et al., 2024). This occurs because CFG presupposes a global guidance scale for all causal parents (e.g. attributes) regardless of whether they ought to be invariant to particular interventions, leading to increased spurious effects in the prediction.

While Xia et al. (2024) observed attribute amplification in previous models due to counterfactual fine-tuning (Ribeiro et al., 2023), we find that a similar failure mode arises in diffusion models due to the indiscriminate application of global guidance scales to increase intervention effectiveness (Monteiro et al., 2023). This behaviour not only violates the underlying causal graph by modifying attributes outside the causal pathway, but can also cause the generation trajectory to drift from the original data manifold, degrading identity preservation (Mokady et al., 2023). Thus, we argue that addressing CFG-induced attribute amplification is critical for its reliable use in counterfactual inference models.

To address the spurious effects of CFG under causal interventions, we propose *Decoupled Classifier-Free Guidance* (DCFG), a general inference-time guidance technique that significantly reduces spurious attribute amplification, without requiring any changes to the underlying diffusion model. DCFG can be implemented via a simple attribute-split embedding strategy that disentangles semantic attributes in the embedding space, and enables selective masking and group-wise modulation at inference time following a causal graph. Unlike standard CFG, DCFG assigns separate weights to attribute groups, allowing for fine-grained, interpretable control over the generative process. While conceptually related to compositional diffusion approaches, our method differs significantly: Shen et al. (2024) apply pixel-wise spatial masks to modulate guidance locally, and Liu et al. (2022) rely on multiple conditional diffusion models fused via shared score functions. In contrast, DCFG uses a single model and modulates guidance at the semantic attribute level. For counterfactual generation, we instantiate DCFG by grouping attributes according to their causal roles (e.g., *intervened* vs. *invariant*) and applying distinct guidance to each group. Crucially, by decoupling guidance and focusing it solely on the intended intervention, DCFG reduces the risk of the generation trajectory drifting away from the original data manifold (Yang et al., 2023; Mokady et al., 2023; Tang et al., 2024). The DCFG framework is general and supports arbitrary partitions of semantic attributes under reasonable independence assumptions. In summary, the contributions of this paper are the following:

(i) We identify and analyze the problem of *attribute amplification* in standard classifier-free guidance, where a global guidance weight causes spurious changes to non-intervened attributes;

(ii) We propose *Decoupled Classifier-Free Guidance* (DCFG), a simple, flexible, and model-agnostic extension of CFG that assigns separate guidance weights to attribute groups and supports arbitrary groupings at inference time under mild independence assumptions;

(iii) Through extensive experiments on challenging real-world data (including medical imaging), we show that DCFG mitigates unintended spurious effects, enhances intervention effectiveness, and improves counterfactual reversibility, resulting in more faithful counterfactual generation.

## 2 BACKGROUND

**Structural Causal Models.** SCMs (Pearl, 2009) consist of a triplet $\langle U, A, F \rangle$, where $U = \{u_i\}_{i=1}^{K}$ denotes the set of exogenous (latent) variables, $A = \{a_i\}_{i=1}^{K}$ the set of endogenous (observed) variables, and $F = \{f_i\}_{i=1}^{K}$ a collection of structural assignments such that each variable $a_k$ is determined by a function $f_k$ of its parents $\mathbf{pa}_k \subseteq A \setminus a_k$ and its corresponding noise $u_k$, such that $a_k := f_k(\mathbf{pa}_k, u_k)$. SCMs enable causal reasoning and interventions via the *do*-operator, e.g., setting a variable $a_k$ to a fixed value $c$ through $do(a_k := c)$. In this work, we focus on generating image counterfactuals and implement the underlying image synthesis mechanism using diffusion models.

**Counterfactual inference.** A counterfactual represents a *'what-if'* scenario given observed events. We denote an image by $\mathbf{x}$, which is generated via a structural assignment $\mathbf{x} := f(\mathbf{u}, \mathbf{pa})$, given its causal parents $\mathbf{pa}$ and exogenous noise variable $\mathbf{u}$. Counterfactual inference (Pearl, 2009) proceeds in three steps: (i) *Abduction:* infer the latent noise $\mathbf{u}$ from the observed data and its parents, i.e. $\mathbf{u} = f^{-1}(\mathbf{x}, \mathbf{pa})$; (ii) *Action:* apply an intervention to alter selected parent variables, yielding the counterfactual parents $\widetilde{\mathbf{pa}}$; (iii) *Prediction:* propagate the effect of the intervention through the model to compute a counterfactual as follows: $\tilde{\mathbf{x}} = f(f^{-1}(\mathbf{x}, \mathbf{pa}), \widetilde{\mathbf{pa}})$. Recent advancements have sought to implement these steps using deep generative model components, such as normalizing flows (Pawlowski et al., 2020), VAEs (Ribeiro et al., 2023; Pawlowski et al., 2020; Monteiro et al., 2023), and diffusion models (Sanchez & Tsaftaris, 2022; Komanduri et al., 2024; Rasal et al., 2025). The general idea is to model each structural assignment $f_\theta$ and its inverse $f_\phi^{-1}$ using deep generative models with trainable parameters $\{\theta, \phi\}$. For invertible models such as flows, $\theta$ and $\phi$ coincide.

## 2.1 DIFFUSION MODELS FOR COUNTERFACTUAL INFERENCE

Diffusion models (DMs) (Sohl-Dickstein et al., 2015b; Ho et al., 2020) are latent variable models designed to generate data by gradually removing Gaussian noise from $\mathbf{x}_T \sim \mathcal{N}(\mathbf{0}, \mathbf{I})$ over $T$ steps. Given a clean data sample $\mathbf{x}_0 \sim p_{\text{data}}$, the forward noising process is defined as follows:

$$\mathbf{x}_t = \sqrt{\alpha_t}\,\mathbf{x}_0 + \sqrt{1 - \alpha_t}\,\boldsymbol{\epsilon}, \qquad \boldsymbol{\epsilon} \sim \mathcal{N}(\mathbf{0}, \mathbf{I}), \tag{1}$$

where $\{\alpha_t\}_{t=0}^T$ is a chosen noise schedule with $\alpha_t \in (0, 1]$, $\alpha_0 = 1$ and $\alpha_T \approx 0$. To learn the reverse process, a parameterized network $\boldsymbol{\epsilon}_\theta(\mathbf{x}_t, t, \mathbf{c})$ is trained to predict the added noise from noisy inputs. We adopt the conditional diffusion model formulation, where $\mathbf{c}$ denotes an embedding of semantic parent attributes $\mathbf{pa}$ used as conditioning. The training objective minimizes the noise prediction loss:

$$\min_\theta \mathbb{E}_{\mathbf{x}_0 \sim p_{\text{data}}, \boldsymbol{\epsilon} \sim \mathcal{N}(\mathbf{0}, \mathbf{I}), t \sim \text{Unif}(\{1, \ldots, T\})} \left[ \|\boldsymbol{\epsilon} - \boldsymbol{\epsilon}_\theta(\mathbf{x}_t, t, \mathbf{c})\|_2^2 \right]. \tag{2}$$

At inference time, data samples are generated by progressively denoising $\mathbf{x}_T$ from time $T$ to time 0. Following Song et al. (2020), the denoising step from $\mathbf{x}_t$ to $\mathbf{x}_{t-1}$ is given by the formula:

$$\mathbf{x}_{t-1} = \sqrt{\alpha_{t-1}} \left( \frac{\mathbf{x}_t - \sqrt{1 - \alpha_t}\,\boldsymbol{\epsilon}_\theta(\mathbf{x}_t, t, \mathbf{c})}{\sqrt{\alpha_t}} \right) + \sqrt{1 - \alpha_{t-1} - \sigma_t^2}\,\boldsymbol{\epsilon}_\theta(\mathbf{x}_t, t, \mathbf{c}) + \sigma_t \boldsymbol{\epsilon}_t, \tag{3}$$

where $\boldsymbol{\epsilon}_t \sim \mathcal{N}(\mathbf{0}, \mathbf{I})$. Setting $\sigma_t = 0$ yields a deterministic sampling process known as DDIM (Song et al., 2020), which defines an invertible trajectory between data and latent space. Following Sanchez & Tsaftaris (2022); Sanchez et al. (2022a); Fontanella et al. (2024); Pérez-García et al. (2024); Rasal et al. (2025), we adopt this DDIM formulation for counterfactual generation, as detailed below.

**Abduction.** We implement the abduction function $\mathbf{u} = f_\theta^{-1}(\mathbf{x}_0, \mathbf{pa})$ using the DDIM forward trajectory. Given an observed image $\mathbf{x}_0$ and conditioning $\mathbf{c}$ representing an embedding vector of semantic parents $\mathbf{pa}$, the latent $\mathbf{x}_T$ serves as a deterministic estimate of the exogenous noise $\mathbf{u}$:

$$\mathbf{x}_{t+1} = \sqrt{\alpha_{t+1}}\hat{\mathbf{x}}_0 + \sqrt{1 - \alpha_{t+1}}\boldsymbol{\epsilon}_\theta(\mathbf{x}_t, t, \mathbf{c}), \quad \hat{\mathbf{x}}_0 = \frac{1}{\sqrt{\alpha_t}} \left( \mathbf{x}_t - \sqrt{1 - \alpha_t}\boldsymbol{\epsilon}_\theta(\mathbf{x}_t, t, \mathbf{c}) \right), \tag{4}$$

for $t = 0, \ldots, T-1$, where $\hat{\mathbf{x}}_0$ is the model's estimate of the clean image at each time $t$.

**Action.** We apply an intervention to the semantic attributes $\mathbf{pa}$ (e.g., do(Male = 1)), and propagate the effect through the causal graph using invertible flows as in (Pawlowski et al., 2020; Ribeiro et al., 2023). This yields the counterfactual attribute vector $\widetilde{\mathbf{pa}}$ and its embedding $\tilde{\mathbf{c}}$.

**Prediction.** We implement the structural assignment $\tilde{\mathbf{x}} := f_\theta(\mathbf{u}, \widetilde{\mathbf{pa}})$ under the modified condition $\tilde{\mathbf{c}}$ using the DDIM reverse trajectory, with $\mathbf{u} = \mathbf{x}_T$ the exogenous noise estimated in eq. (4):

$$\mathbf{x}_{t-1} = \sqrt{\alpha_{t-1}}\hat{\mathbf{x}}_0 + \sqrt{1 - \alpha_{t-1}}\boldsymbol{\epsilon}_\theta(\mathbf{x}_t, t, \tilde{\mathbf{c}}), \qquad \text{for} \qquad t = T, \ldots, 1, \tag{5}$$

where $\hat{\mathbf{x}}_0$ is the predicted clean image, and the final output $\tilde{\mathbf{x}}_0$ is the predicted counterfactual $\tilde{\mathbf{x}}$. In practice, decoding under the counterfactual condition $\tilde{\mathbf{c}}$ using the conditional denoiser alone may be insufficient for producing effective counterfactuals. Additional guidance is often required to steer generation toward the desired intervention (Sanchez & Tsaftaris, 2022; Sanchez et al., 2022a; Komanduri et al., 2024; Fontanella et al., 2024; Weng et al., 2024; Song et al., 2024; Pérez-García et al., 2024; Rasal et al., 2025; Kumar et al., 2025). In line with previous work, we adopt classifier-free guidance (CFG) to enhance counterfactual fidelity and alignment with the specified intervention.

## 2.2 CLASSIFIER-FREE GUIDANCE

Classifier-free guidance (Ho & Salimans, 2022) is a widely adopted technique in conditional diffusion models. It enables conditional generation without requiring an external classifier by training a single denoising model to operate in both conditional and unconditional modes. During training, the model learns both $p_\theta(\mathbf{x}_t \mid \mathbf{c})$ and $p_\theta(\mathbf{x}_t \mid \varnothing)$ by randomly replacing $\mathbf{c}$ with a null token $\varnothing$. At inference time, CFG biases the sampling process toward regions more consistent with the conditioning signal, which can be understood as sampling from a reweighted conditional distribution of the form:

$$p^\omega(\mathbf{x}_t \mid \mathbf{c}) \propto p(\mathbf{x}_t)p(\mathbf{c} \mid \mathbf{x}_t)^\omega, \tag{6}$$

where $\omega \geq 0$ controls the guidance strength. This corresponds to interpolating between the unconditional and conditional scores:

$$\nabla_{\mathbf{x}_t} \log p^\omega(\mathbf{x}_t \mid \mathbf{c}) = (1 - \omega) \cdot \nabla \log p(\mathbf{x}_t) + \omega \cdot \nabla \log p(\mathbf{x}_t \mid \mathbf{c}). \tag{7}$$

In practice, this is implemented by combining the model's predictions with and without conditioning:

$$\boldsymbol{\epsilon}_{\mathrm{CFG}}(\mathbf{x}_t, t, \mathbf{c}) = \boldsymbol{\epsilon}_\theta(\mathbf{x}_t, t, \varnothing) + \omega \cdot (\boldsymbol{\epsilon}_\theta(\mathbf{x}_t, t, \mathbf{c}) - \boldsymbol{\epsilon}_\theta(\mathbf{x}_t, t, \varnothing)). \tag{8}$$

With CFG, *abduction* is the same as in eq. (4), and *action* remains unchanged. The only difference lies in the *prediction* step, where the conditional denoiser is replaced with the guided score $\boldsymbol{\epsilon}_{\mathrm{CFG}}(\mathbf{x}_t, t, \tilde{\mathbf{c}})$ to enhance counterfactual effectiveness (Sanchez et al., 2022a; Komanduri et al., 2024):

$$\mathbf{x}_{t-1} = \frac{\sqrt{\alpha_{t-1}}}{\sqrt{\alpha_t}} \left( \mathbf{x}_t - \sqrt{1 - \alpha_t} \boldsymbol{\epsilon}_{\mathrm{CFG}}(\mathbf{x}_t, t, \tilde{\mathbf{c}}) \right) + \sqrt{1 - \alpha_{t-1}} \boldsymbol{\epsilon}_{\mathrm{CFG}}(\mathbf{x}_t, t, \tilde{\mathbf{c}}). \tag{9}$$

Despite its effectiveness, CFG applies a single global guidance weight $\omega$ uniformly across the entire counterfactual embedding $\tilde{\mathbf{c}}$, which typically encodes multiple attributes, including some that may not be altered by particular interventions. In counterfactual generation, however, only a subset of attributes in $\tilde{\mathbf{c}}$ (i.e., those affected by the intervention) should be emphasized, while the remaining attributes should remain invariant. Applying the same guidance strength to all elements of $\tilde{\mathbf{c}}$ violates this principle, and can cause unintended changes to invariant attributes. This misalignment is called *attribute amplification* (Xia et al., 2024), which violates the relationships in the associated causal graph, undermining the axiomatic soundness of inferred counterfactuals (Monteiro et al., 2023).

To address the limitations of CFG for counterfactual inference, we propose a structured alternative that assigns separate guidance weights to semantically or causally defined groups of attributes.

## 3 DECOUPLED CLASSIFIER-FREE GUIDANCE

In this section, we present our *Decoupled Classifier-Free Guidance* (DCFG) for counterfactual image generation. We first propose a simple *attribute-split conditioning embedder* (section 3.1) as a practical implementation that separates attributes in the embedding space to enable selective control. Building on this, we then describe our DCFG formulation in detail (section 3.2), which allows distinct guidance strengths to be applied to different subsets of attributes within an assumed causal graph. Finally, we present how DCFG is integrated into DDIM-based counterfactual inference (section 3.3), detailing its application across abduction, action, and prediction steps using causally defined attribute groupings.

### 3.1 ATTRIBUTE-SPLIT CONDITIONING EMBEDDING

In practice, raw conditioning inputs such as discrete image labels or structured attributes (e.g., a patient's *sex*, *race*, or *disease status*) are not used directly in diffusion models but transformed into dense vectors using embedding functions, typically via multi-layer perceptrons (MLPs) (Dhariwal & Nichol, 2021), convolutional encoders (Zhang et al., 2023), or transformer-based text encoders (Ho & Salimans, 2022; Ramesh et al., 2022). These embeddings align semantic or categorical inputs with the model's internal representations, but conventional designs often entangle multiple attributes into a single conditioning vector, making it difficult to independently control attributes during sampling.

To address this, we introduce a simple *attribute-split conditioning embedding* technique that preserves the identity of each attribute in the embedding space. Let $pa_i$ denote the raw value of the $i$-th parent attribute (e.g., a binary indicator or scalar). Each $pa_i$ is embedded independently via a dedicated MLP: $\mathcal{E}_i : \mathbb{R}^{d_i} \to \mathbb{R}^d$, and the final condition vector is formed by concatenating the outputs:

$$\mathbf{c} = \mathrm{concat}\left( \mathcal{E}_1(pa_1), \mathcal{E}_2(pa_2), \ldots, \mathcal{E}_K(pa_K) \right), \qquad \text{where} \qquad \mathbf{c} \in \mathbb{R}^{Kd}. \tag{10}$$

This architecture provides a flexible representation where each attribute is explicitly disentangled at the embedding level. As a result, we can selectively null-tokenize or modulate individual attributes at inference time, enabling fine-grained control. Throughout the rest of the paper, we denote the semantic attribute vector as $\mathbf{pa}$ and the corresponding embedding vector as $\mathbf{c}$, as defined in eq. (10).

## 3.2 FORMULATION: GROUP-WISE DCFG

To overcome the limitations of CFG and enable more precise, causally aligned control in counterfactual image generation, we propose *Decoupled Classifier-Free Guidance* (DCFG). Rather than applying a single guidance weight uniformly to the entire conditioning vector, we partition semantic attributes $\mathbf{pa}$ into $M$ disjoint groups $\mathbf{pa}^{(1)}, \mathbf{pa}^{(2)}, \ldots, \mathbf{pa}^{(M)}$, and apply a separate guidance weight $\omega_m$ to each group. Let $\mathbf{pa} = (pa_1, \ldots, pa_K)$ denote the vector of semantic parent attributes.

**Proposition 1** (Proxy Posterior for DCFG). *Under the assumption that the groups $\mathbf{pa}^{(1)}, \ldots, \mathbf{pa}^{(M)}$ are conditionally independent given the latent variable $\mathbf{x}_t$, for any time $t$, that is: $p(\mathbf{pa} \mid \mathbf{x}_t) = \prod_{m=1}^{M} p(\mathbf{pa}^{(m)} \mid \mathbf{x}_t)$, we obtain the following factorized proxy posterior:*

$$p^\omega(\mathbf{x}_t \mid \mathbf{pa}) \propto p(\mathbf{x}_t) \prod_{m=1}^{M} p(\mathbf{pa}^{(m)} \mid \mathbf{x}_t)^{\omega_m}, \tag{11}$$

*where $\omega_m \geq 0$ controls the guidance strength for each group $m$.*

A complete derivation and score-based justification for this proxy posterior is provided in Appendix B. The corresponding guided update used in score-based diffusion sampling is then given by:

$$\nabla_{\mathbf{x}_t} \log p^\omega(\mathbf{x}_t \mid \mathbf{pa}) = \nabla \log p(\mathbf{x}_t) + \sum_{m=1}^{M} \omega_m \cdot \left( \nabla \log p(\mathbf{x}_t \mid \mathbf{pa}^{(m)}) - \nabla \log p(\mathbf{x}_t) \right). \tag{12}$$

In practice, we encode $\mathbf{pa}$ into a dense conditioning vector $\mathbf{c}$ using the attribute-split embedding described in Section 3.1. For each group $m$, we construct a masked embedding $\underline{\mathbf{c}}^{(m)}$ that retains only the embeddings for $\mathbf{pa}^{(m)}$ and replaces all others with null tokens (represented here as zero vectors):

$$\underline{\mathbf{c}}^{(m)} = \text{concat}\left( \delta_1^{(m)} \cdot \mathcal{E}_1(pa_1), \ldots, \delta_K^{(m)} \cdot \mathcal{E}_K(pa_K) \right), \quad \delta_i^{(m)} = \begin{cases} 1, & \text{if } pa_i \in \mathbf{pa}^{(m)} \\ 0, & \text{otherwise} \end{cases} \tag{13}$$

The final guided score used in the diffusion model is computed as follows:

$$\boldsymbol{\epsilon}_{\text{DCFG}}(\mathbf{x}_t, t, \mathbf{c}) = \boldsymbol{\epsilon}_\theta(\mathbf{x}_t, t, \varnothing) + \sum_{m=1}^{M} \omega_m \cdot \left( \boldsymbol{\epsilon}_\theta(\mathbf{x}_t, t, \underline{\mathbf{c}}^{(m)}) - \boldsymbol{\epsilon}_\theta(\mathbf{x}_t, t, \varnothing) \right). \tag{14}$$

The proposed DCFG framework is highly flexible, as it allows arbitrary groupings of attributes, regardless of whether attributes within a group are mutually independent or not. The only assumption required is that different groups are conditionally independent given the latent variable $\mathbf{x}_t$. This flexibility enables a wide range of configurations. For instance, setting $M = 1$ recovers standard global CFG, while increasing $M$ provides progressively finer-grained control, including per-attribute guidance ($M = K$) as an extreme case where we assume all attributes are independent of each other.

## 3.3 DCFG FOR COUNTERFACTUAL GENERATION

We now detail how DCFG is straightforwardly integrated into DDIM-based counterfactual inference.

**Abduction.** The abduction step proceeds as in eq. (4), where the conditioning vector $\mathbf{c}$ is obtained by embedding the semantic parent attributes $\mathbf{pa}$ using the attribute-split encoder defined in eq. (10).

**Action.** As in previous setups, we apply a causal intervention to obtain a modified semantic vector $\widetilde{\mathbf{pa}}$. This is then embedded into the counterfactual conditioning vector $\tilde{\mathbf{c}}$ via the attribute-split embedder:

$$\tilde{\mathbf{c}} = \text{concat}\left( \mathcal{E}_1(\widetilde{pa}_1), \ldots, \mathcal{E}_K(\widetilde{pa}_K) \right). \tag{15}$$

**Prediction.** The prediction step uses the DCFG-guided reverse DDIM trajectory:

$$\mathbf{x}_{t-1} = \sqrt{\alpha_{t-1}} \hat{\mathbf{x}}_0 + \sqrt{1 - \alpha_{t-1}} \boldsymbol{\epsilon}_{\text{DCFG}}(\mathbf{x}_t, t, \tilde{\mathbf{c}}), \tag{16}$$

$$\text{where} \quad \hat{\mathbf{x}}_0 = \frac{1}{\sqrt{\alpha_t}} \left( \mathbf{x}_t - \sqrt{1 - \alpha_t} \boldsymbol{\epsilon}_{\text{DCFG}}(\mathbf{x}_t, t, \tilde{\mathbf{c}}) \right), \tag{17}$$

and $\boldsymbol{\epsilon}_{\text{DCFG}}(\mathbf{x}_t, t, \tilde{\mathbf{c}})$ is computed as in eq. (14) using counterfactual conditioning embedding $\tilde{\mathbf{c}}$.

For counterfactual generation, we can adopt a two-group partitioning of attributes based on the assumed causal graph. The *affected group* $\mathbf{pa}^{\text{aff}}$ contains attributes directly intervened upon and their descendants, while the *invariant group* $\mathbf{pa}^{\text{inv}}$ comprises attributes expected to remain unchanged. These groups are assumed conditionally independent given the latent $\mathbf{x}_t$, consistent with the d-separation of the post-intervention graph. Under this setup, eq. (14) uses $M = 2$ groups with separate guidance weights $\omega_{\text{aff}}$ and $\omega_{\text{inv}}$. Note that the two-group partition is only one possible choice: guidance can also be separated at finer-grained level, including the attribute level, provided conditional independence holds. We present such extensions, including multi-attribute interventions and attribute-wise configurations, in Section G, which further demonstrate the generality and flexibility of DCFG.

## 4 EXPERIMENTS

In this section, we demonstrate the benefits of the proposed approach across three public datasets. For each dataset, we train a diffusion model with the same architecture and training protocol, detailed in section C. We compare our DCFG against the standard CFG baseline. In all results, settings labeled as $\omega = X$ correspond to standard classifier-free guidance (CFG) with a global guidance weight. In contrast, configurations denoted by $\omega_{\text{aff}}{=}X$, $\omega_{\text{inv}}{=}Y$ represent the two-group DCFG, where separate guidance weights are applied to the intervened and invariant attribute groups, respectively. Following Monteiro et al. (2023); Melistas et al. (2024), we evaluate counterfactual quality using two metrics. **Effectiveness** ($\Delta$): Measured by a pretrained classifier as the change in AUROC for intervened attributes relative to $\omega = 1.0$ (no CFG). Higher $\Delta$ indicates stronger intervention effect; large $\Delta$ on invariant attributes indicates unintended amplification. **Reversibility**: Assesses how well counterfactuals can be reversed to the original image using inverse interventions. We report MAE and LPIPS; lower values indicate better identity preservation. See section A.2 for details.

### 4.1 CASE STUDY 1: CELEBA

We begin our empirical evaluation of DCFG on the CelebA-HQ dataset (Karras et al., 2017), using `Smiling`, `Male`, and `Young` as independent binary attributes. We adopt this simplified setup to isolate inference-time failures of standard CFG in a controlled setting. Under this designed scenario, unintended changes in non-intervened attributes can be attributed to attribute amplification rather than valid causal effects. We first evaluate DCFG under single-attribute interventions, then extend to multi-attribute settings to highlight its flexibility. Refer to section D.1 for more dataset details.

Fig. 1 presents the $\Delta$ metrics under different guidance strategies for two separate interventions, namely `do(Smiling)` and `do(Young)`. As the global guidance weight $\omega$ of CFG increases (left to right side of each plot), the $\Delta$ of the intervened attribute improves, but so do the $\Delta$ values of attributes that should remain invariant, indicating undesirable spurious amplification. In contrast, the right side of each plot shows results for DCFG, where distinct weights are applied to affected ($\omega_{\text{aff}}$) and invariant ($\omega_{\text{inv}}$) attribute groups. This decoupled formulation achieves comparable or stronger improvement on the intervened attribute while keeping the others stable, validating the ability of DCFG to produce more disentangled and effective counterfactuals purely at inference time.

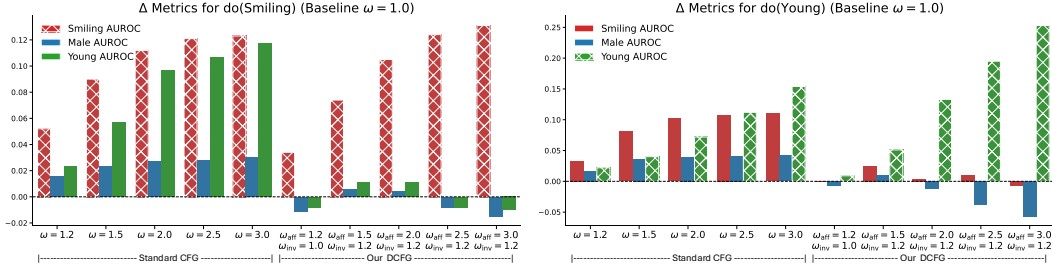

Figure 1: Comparison of $\Delta$ metrics under different interventions in CelebA-HQ. Left: Intervention on `Smiling`. Right: Intervention on `Young`. Both use baseline $\omega{=}1.0$. Under global CFG, increasing $\omega$ boosts the intended attribute but amplifies non-target ones. DCFG achieves similar improvements on the target attribute while mitigating amplification. See section D.2 for full quantitative results.

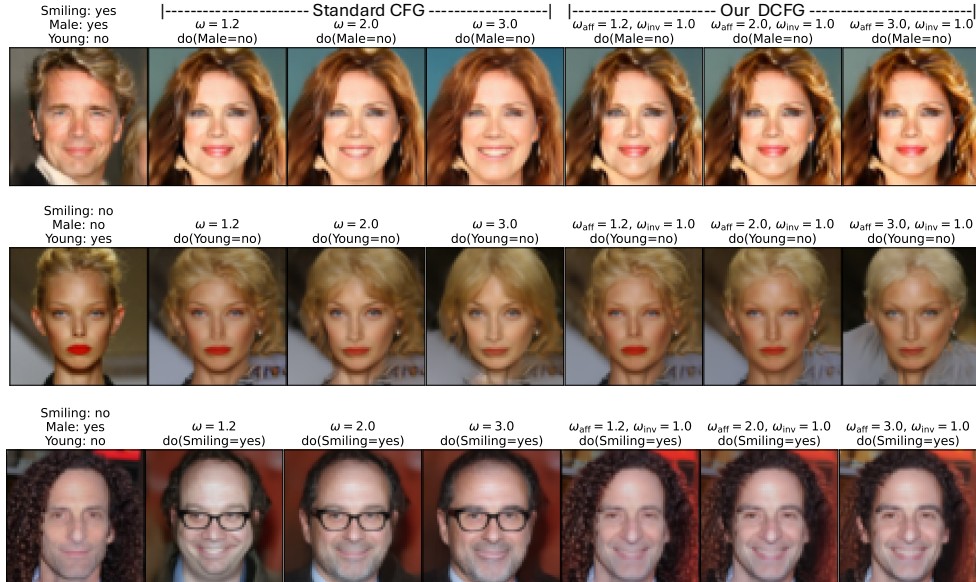

Figure 2: Counterfactual generations in CelebA-HQ ($64 \times 64$). Each row compares global CFG (left) and DCFG (right) across guidance weights. Top: global CFG causes amplification of `Smiling` under `do(Male)`; Middle: `do(Young)` suppresses `Male` (i.e. amplifies `Male=`*no*); Bottom: `do(Smiling)` makes the subject appear older, adds glasses, and alters identity. DCFG mitigates these unintended changes and preserves *invariant* attributes. See section D.3 for more visual results.

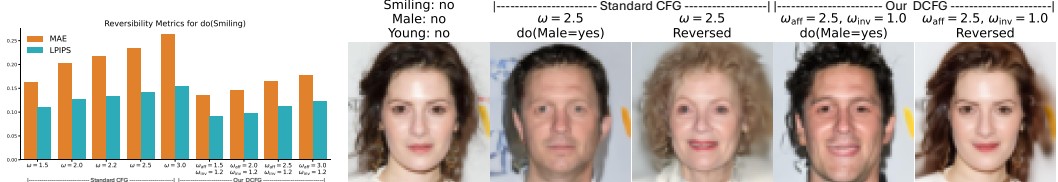

Figure 3: Reversibility analysis in CelebA-HQ ($64 \times 64$). Left: Quantitative evaluation of how well the original image is recovered after generating a counterfactual and mapping it back to the original condition under `do(Smiling)`. Right: A qualitative example showing a counterfactual generated under `do(Male)` and its reconstruction after reversing the intervention with CFG and our DCFG.

Fig. 2 illustrates how global CFG can introduce unitended changes by uniformly amplifying all conditioning signals, even when only one attribute is meant to change. In the top row, applying `do(Male=no)` with increasing $\omega$ inadvertently amplifies `Smiling`; in the middle row, `do(Young=no)` reduces `Male` expression; and in the bottom row, `do(Smiling=yes)` introduces changes to age, identity, and even adds glasses. These unintended shifts stem from global CFG treating all attributes equally. In contrast, DCFG applies decoupled guidance across attributes, assigning stronger weights to those affected by the intervention, allowing attributes that were not targeted by the intervention to remain unchanged. This results in counterfactuals that more faithfully reflect the intended change while preserving identity and consistency in non-intervened factors.

Fig. 3 evaluates the reversibility of counterfactuals in CelebA-HQ. The left panel shows MAE and LPIPS when recovering the original after applying and then reversing an intervention (e.g., `do(Smiling)`). With global CFG, errors grow as guidance strength increases, while DCFG yields consistently lower values for the same settings, improving recovery. The right panel shows a qualitative example under `do(Male)`, where global CFG amplifies non-intervened attributes (e.g., `Young`), making the reversed image appear older. In contrast, DCFG applies strong guidance only to intervened attributes, mitigating amplification and producing more faithful, disentangled, and reversible counterfactuals. More reversibility results are provided in sections D.2 and D.3.

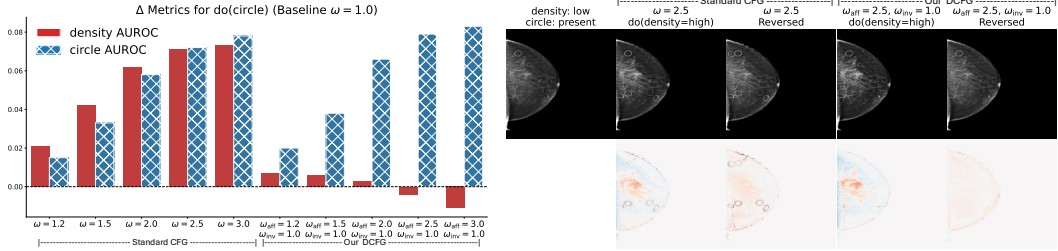

Figure 4: Qualitative results for `do(Smiling, Male, Young)`. We compare two-group DCFG ($\omega_{\text{aff}} = 2.5, \omega_{\text{inv}} = 1.0$) with attribute-wise DCFG, where $\omega_s$, $\omega_m$, and $\omega_y$ control guidance for `Smiling`, `Male`, and `Young`. Symmetric weights ($\omega_s = \omega_m = \omega_y = 2.5$) reproduce two-group results, while asymmetric weights highlight DCFG's flexibility. See section G.2 for more results.

Finally, to illustrate that DCFG extends beyond the two-group setting, we consider a three-attribute intervention `do(Smiling, Male, Young)`. As shown in Fig. 4, we compare the two-group formulation ($\omega_{\text{aff}} = 2.5, \omega_{\text{inv}} = 1.0$) with attribute-wise DCFG, where each attribute has its own weight ($\omega_s, \omega_m, \omega_y$). Symmetric weights ($\omega_s = \omega_m = \omega_y = 2.5$) recover the two-group results, while asymmetric settings demonstrate the additional flexibility to selectively emphasize individual attributes. Further discussion and results on multi-attribute interventions are provided in section G.

## 4.2 CASE STUDY 2: MAMMOGRAPHY

In this study, we evaluate DCFG on the EMBED (Jeong et al., 2023) breast mammography dataset. For details about EMBED, the reader may refer to section E.1. For our experiments, we define a binary `circle` attribute based on the presence of circular skin markers, and a binary breast `density` label, where categories A and B are grouped as `low` and categories C and D as `high`.

Fig. 5 presents results for counterfactual generation on EMBED. The bar plot on the left reports $\Delta$ effectiveness metrics, measuring how classifier performance changes relative to the baseline. While global CFG improves effectiveness for the target attribute (`circle`), it also increases effectiveness on non-intervened attributes such as `density`, indicating unintended attribute amplfication. DCFG mitigates this by applying selective guidance, maintaining stable performance on non-target attributes. The figure on the right illustrates a key example: applying `do(density)` under global CFG unintentionally amplifies the presence of circular skin markers, as evidenced by the increased number of visible circles in both the counterfactual and reversed images. This is suppressed under DCFG, where `circle` features remain unchanged in both counterfactual and reversed images.

## 4.3 CASE STUDY 3: CHEST RADIOGRAPHY

We evaluate our method on the MIMIC-CXR dataset (Johnson et al., 2019). We follow the dataset splits and filtering protocols from Ribeiro et al. (2023), and focus on the binary disease label of pleural effusion. The underlying causal graph in Ribeiro et al. (2023) includes four attributes: `race`, `sex`,

Figure 5: Evaluation of counterfactual generation on EMBED ($192 \times 192$). Left: $\Delta$ metrics showing the effect of `do(circle)`. DCFG improves target intervention effectiveness while suppressing spurious shifts in non-intervened attributes. Right: A visual example showing the input image, the counterfactual under `do(density)`, the reversed image, and their difference maps (CF/Rev. - input). See sec. E.2 for full quantitative results and sec. E.3 for more visual results.

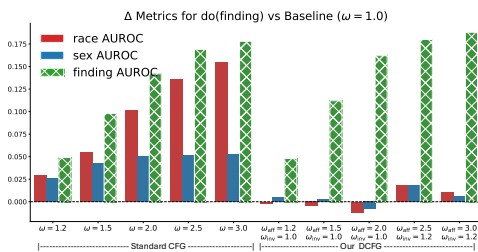 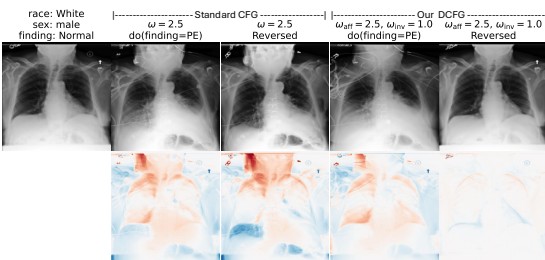

Figure 6: Evaluation of counterfactual generation on MIMIC (192 × 192). Left: Δ metrics showing the effect of `do(finding)`. DCFG improves target intervention effectiveness while suppressing spurious shifts in non-intervened attributes. Right: A visual example showing the input image, the counterfactual under `do(density)`, the reversed image, and their difference maps (CF/Rev. - input). See section F.2 for full quantitative results and section F.3 for more qualitative results.

`finding`, and `age`. We adopt this setup, but since our goal is to study *attribute amplification* caused by CFG, we focus on `sex`, `race`, and `finding`, which we assume to be mutually independent for the purposes of our analysis. The reader may refer to section F.1 for further details.

Fig. 6 presents an evaluation of counterfactual generation in MIMIC-CXR, highlighting the advantages of our proposed DCFG. The bar plot on the left shows Δ metrics that quantify the change in effectiveness relative to the baseline $\omega$=1.0. While global CFG improves effectiveness for the intervened variable (`finding`) as expected, it also introduces substantial spurious shifts in non-intervened attributes such as `race` and `sex`, revealing unwanted attribute amplification. In contrast, DCFG achieves comparable or higher intervened effectiveness while suppressing spurious amplification, demonstrating more precise and controlled generation. On the right, we show a qualitative example of a counterfactual generated under `do(finding)`, its reversed reconstruction, and their corresponding difference maps. Compared to global CFG, our method yields localized, clinically meaningful changes in counterfactuals and better identity preservation in the reversed image.

## 5 CONCLUSION

In this work, we identify and address a key limitation of classifier-free guidance for counterfactual image generation: the application of a global uniform guidance scale for all conditioning attributes leads to spurious amplification of factors that should remain unchanged under causal interventions. To address this, we proposed *Decoupled Classifier-Free Guidance* (DCFG), a new flexible guidance technique that allows arbitrary grouping of semantic attributes, with distinct guidance weights applied to each group under the mild assumption of conditional independence. DCFG is primarily operationalized via a simple two-group partition into *intervened* and *invariant* attributes, but also supports more fine-grained settings, such as multi-attribute and per-attribute configurations following a prescribed causal graph. This flexibility enables DCFG to suppress spurious changes outside of the intervention's causal pathway while preserving the intended effect. Beyond counterfactuals, DCFG can apply broadly to conditional generation tasks that benefit from group-wise control over conditioning signals. We evaluated DCFG on CelebA-HQ, EMBED, and MIMIC-CXR, covering natural and medical image domains. Our results show that DCFG significantly reduces attribute amplification while maintaining intervention effectiveness with improved identity preservation, particularly at higher guidance strengths. However, the selection of guidance weights still requires empirical tuning. Future work could explore learned strategies to adaptively tune these weights based on the input condition or diffusion timestep. Future work could explore learned strategies to adaptively tune these weights based on the input condition or diffusion timestep. Additionally, DCFG's modular nature opens up opportunities for broader integration. One direction is combining DCFG with latent diffusion models or diffusion autoencoders to enable high-resolution synthesis and stronger identity preservation. Another promising extension is to apply DCFG selectively at specific diffusion timesteps. Prior work suggests that restricting guidance to mid-to-late timesteps can mitigate over-saturation and improve generation quality. Exploring such selective or dynamic scheduling strategies within the DCFG framework may further enhance counterfactual fidelity.

**Ethics statement.** This work uses publicly available datasets, including CelebA-HQ (human faces), MIMIC-CXR (chest X-rays), and EMBED (mammography). All datasets were released with appropriate ethical approvals and consent processes in place by their providers. We use them solely for research purposes, without attempting to identify individuals or deploy the models in clinical or biometric applications. While counterfactual image generation has potential for misuse (e.g., in manipulating facial attributes), our focus is on scientific study of causal generative modeling.

**Reproducibility statement.** We provide dataset descriptions in sections D.1, E.1 and F.1, model architectures and training hyperparameters in section C, and evaluation protocols in sections A.2 and 4. Code will be made publicly available upon acceptance.

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

# A BACKGROUND

## A.1 NOTATION SUMMARY

| Symbol | Description |
|---|---|
| $\mathbf{x}_0$ (also denoted as $\mathbf{x}$) | Original observed image |
| $\mathbf{x}_t$ | Noisy image at diffusion timestep $t$ |
| $\mathbf{x}_T$ (i.e., $\mathbf{u}$) | Latent code after DDIM forward process (abduction; exogenous noise) |
| $\tilde{\mathbf{x}}$ | Generated counterfactual image |
| $\epsilon_\theta(\mathbf{x}_t, t, \cdot)$ | Denoiser prediction given condition input |
| $\mathbf{pa}$ | Vector of semantic parent attributes (e.g., *sex*, *age*) |
| $\widetilde{\mathbf{pa}}$ | Counterfactual parent attributes after intervention |
| $pa_i$ | Raw value of the $i$-th semantic attribute |
| $\mathcal{E}_i$ | Embedding MLP for $pa_i$: $\mathbb{R}^{d_i} \to \mathbb{R}^d$ |
| $\mathbf{c}$ | Full conditioning vector from $\mathbf{pa}$ |
| $\tilde{\mathbf{c}}$ | Conditioning vector from counterfactual attributes $\widetilde{\mathbf{pa}}$ |
| $\varnothing$ | Null token for classifier-free guidance (unconditional input) |
| $\omega_m$ | CFG weight for attribute group $m$ |
| $\mathbf{pa}^{(m)}$ | Attributes in the $m$-th group |
| $\underline{\mathbf{c}}^{(m)}$ | Masked condition vector preserving only group $m$ |

Table A.1: Notation used throughout the paper. Tilde ($\sim$) indicates counterfactual quantities.

## A.2 EVALUATING COUNTERFACTUALS

To evaluate the soundness of generated counterfactuals, we define a counterfactual image generation function $\mathcal{F}_\theta(\cdot)$, which produces counterfactuals according to

$$\tilde{\mathbf{x}} := \mathcal{F}_\theta(\mathbf{x}, \mathbf{pa}, \tilde{\mathbf{pa}}) = f_\theta(f_\theta^{-1}(\mathbf{x}, \mathbf{pa}), \tilde{\mathbf{pa}}). \tag{18}$$

We describe three key metrics used to assess counterfactual quality: composition, reversibility, and effectiveness (Monteiro et al., 2023; Melistas et al., 2024).

**Composition** evaluates how well the model reconstructs the original image under a null intervention, by computing a distance metric $d$ between the original image and its counterfactual:

$$\text{Comp}(\mathbf{x}, \mathbf{pa}) := d(\mathbf{x}, \mathcal{F}_\theta(\mathbf{x}, \mathbf{pa}, \mathbf{pa})). \tag{19}$$

**Reversibility** measures the consistency of the counterfactual transformation by applying the reverse intervention and comparing the result to the original image:

$$\text{Rev}(\mathbf{x}, \mathbf{pa}, \tilde{\mathbf{pa}}) := L_1(\mathbf{x}, \mathcal{F}_\theta(\mathcal{F}_\theta(\mathbf{x}, \mathbf{pa}, \tilde{\mathbf{pa}}), \tilde{\mathbf{pa}}, \mathbf{pa})). \tag{20}$$

**Effectiveness** quantifies whether the intended intervention has the desired causal effect. It compares the intervened value $\tilde{\mathbf{pa}}_k$ with the prediction obtained by an anti-causal model applied to the counterfactual image:

$$\text{Eff}(\mathbf{x}, \mathbf{pa}, \tilde{\mathbf{pa}}) := L_k(\tilde{\mathbf{pa}}_k, \mathbf{Pa}_k(\mathcal{F}_\theta(\mathbf{x}, \mathbf{pa}, \tilde{\mathbf{pa}}))). \tag{21}$$

**A note on Composition.** We do not report the **Composition** metric in our evaluation, as it is ill-defined in the context of CFG and DCFG. Since both methods use the same trained diffusion model, applying a null intervention (i.e., $\tilde{\mathbf{pa}} = \mathbf{pa}$) does not meaningfully differentiate between them. If reconstruction is performed without guidance, CFG and DCFG are equivalent, reducing to standard decoding. If guidance is applied, it becomes unclear how to split attributes into invariant ($\mathbf{pa}_{\text{inv}}$) and intervened ($\mathbf{pa}_{\text{aff}}$) groups during null intervention. For instance, assigning all attributes to $\mathbf{pa}_{\text{inv}}$ with $\omega_{\text{inv}} = 1$ would effectively disable guidance, making the comparison trivial and uninformative. For this reason, we focus on **Effectiveness** and **Reversibility**, which better capture the behavior of guided sampling under interventions.

**Effectiveness Classifier.** To evaluate effectiveness, we train a classifier with a ResNet-18 backbone for each dataset, using the split as in table A.2. The classifier predicts the intervened attribute from generated counterfactuals, and AUROC is used to quantify intervention success. On CelebA, the classifier achieves AUROC scores of 0.974 (`Smiling`), 0.992 (`Male`), and 0.828 (`Young`). On EMBED, the AUROC is 0.935 for `density` and 0.908 for `circle`. On MIMIC-CXR, AUROC scores are 0.864 for `race`, 0.991 for `sex`, and 0.938 for `finding`.

**Reversibility Metrics.** We use Mean Absolute Error (MAE) and LPIPS (Zhang et al., 2018) to evaluate reversibility. These metrics quantify the pixel-level and perceptual similarity, respectively, between the original image and its reversed counterfactual.

## B  DECOUPLED CLASSIFIER-FREE GUIDANCE

We provide a theoretical justification for the Decoupled Classifier-Free Guidance formulation presented in Proposition 1, interpreting it as gradient ascent on a sharpened proxy posterior under a group-level conditional independence assumption.

**Proof of Proposition 1.**  We begin by assuming that the semantic attributes are partitioned into $M$ disjoint groups:

$$\mathbf{pa} = (\mathbf{pa}^{(1)}, \ldots, \mathbf{pa}^{(M)}). \tag{22}$$

Under the assumption that these groups are conditionally independent given $\mathbf{x}_t$, we have:

$$p(\mathbf{pa} \mid \mathbf{x}_t) = \prod_{m=1}^{M} p(\mathbf{pa}^{(m)} \mid \mathbf{x}_t). \tag{23}$$

Applying Bayes' rule:

$$p(\mathbf{x}_t \mid \mathbf{pa}) = \frac{p(\mathbf{pa} \mid \mathbf{x}_t) \cdot p(\mathbf{x}_t)}{p(\mathbf{pa})} \propto p(\mathbf{pa} \mid \mathbf{x}_t) \cdot p(\mathbf{x}_t). \tag{24}$$

so the posterior can be factorized as:

$$p(\mathbf{x}_t \mid \mathbf{pa}) \propto p(\mathbf{x}_t) \cdot \prod_{m=1}^{M} p(\mathbf{pa}^{(m)} \mid \mathbf{x}_t). \tag{25}$$

Applying group-level guidance weights $\omega_m$ yields the sharpened proxy posterior:

$$p^{\omega}(\mathbf{x}_t \mid \mathbf{pa}) \propto p(\mathbf{x}_t) \cdot \prod_{m=1}^{M} p(\mathbf{pa}^{(m)} \mid \mathbf{x}_t)^{\omega_m}. \tag{26}$$

**Gradient of the Log Proxy Posterior.**  For DCFG, the gradient becomes:

$$\nabla_{\mathbf{x}_t} \log p^{\omega}(\mathbf{x}_t \mid \mathbf{pa}) = \nabla \log p(\mathbf{x}_t) + \sum_{m=1}^{M} \omega_m \cdot \left( \nabla \log p(\mathbf{x}_t \mid \mathbf{pa}^{(m)}) - \nabla \log p(\mathbf{x}_t) \right), \tag{27}$$

where $\mathbf{pa}^{(m)}$ is the $m$-th group of attributes, and $\omega_m$ is the guidance weight for that group.

The corresponding implementation interpolates denoising scores per group:

$$\boldsymbol{\epsilon}_{\text{CFG}} = \boldsymbol{\epsilon}_\theta(\mathbf{x}_t, t \mid \varnothing) + \sum_{m=1}^{M} \omega_m \cdot \left( \boldsymbol{\epsilon}_\theta(\mathbf{x}_t, t, \underline{\mathbf{c}}^{(m)}) - \boldsymbol{\epsilon}_\theta(\mathbf{x}_t, t, \varnothing) \right), \tag{28}$$

where $\underline{\mathbf{c}}^{(m)}$ denotes the masked condition vector in which only group $m$ is retained and all others are null-tokenized, as defined in eq. (13).

# C  IMPLEMENTATION DETAILS

**Architecture.**  We adopt the commonly used U-net backbone (Dhariwal & Nichol, 2021) for all diffusion models in this work. We modify it to support CFG and DCFG. Each conditioning attribute is projected via a dedicated MLP embedder (section 3.1), and the resulting embeddings are concatenated with the timestep embedding. During training, we apply exponential moving average (EMA) to model weights for improved stability. All images are normalized to the range [-1,1]. The complete architecture and training configurations for each dataset are summarized in table A.2.

Table A.2: Training and architecture of diffusion U-Net configurations used in our experiments. *We did not evaluate on the whole test set due to the high computational cost of diffusion sampling.

| PARAMETER | CELEBA-HQ | EMBED | MIMIC-CXR |
|---|---|---|---|
| TRAIN SET SIZE | 24,000 | 151,948 | 62,336 |
| VALIDATION SET SIZE | 3,000 | 7,156 | 9,968 |
| TEST SET SIZE* | 3,000 | 43,669 | 30,535 |
| RESOLUTION | $64 \times 64 \times 3$ | $192 \times 192 \times 1$ | $192 \times 192 \times 1$ |
| BATCH SIZE | 128 | 48 | 48 |
| TRAINING EPOCHS | 6000 | 5000 | 5000 |
| BASE CHANNELS (U-NET) | 64 | 64 | 32 |
| CHANNEL MULTIPLIERS | [1,2,4,8] | [1,1,2,2,4,4] | [1,2,3,4,5,6] |
| ATTENTION RESOLUTIONS | [16] | - | - |
| RESNET BLOCKS | 2 | 2 | 2 |
| DROPOUT RATE | 0.1 | 0.0 | 0.0 |
| NUM. CONDITIONING ATTRS | 3 | 2 | 4 |
| COND. EMBEDDING DIM | $3 \times 96$ | $2 \times 64$ | $4 \times 64$ |
| NOISE SCHEDULE | LINEAR | COSINE | LINEAR |
| LEARNING RATE | 1e-4 | | |
| OPTIMISER | ADAM (WD 1e-4) | | |
| EMA DECAY | 0.9999 | | |
| TRAINING STEPS $T$ | 1000 | | |
| LOSS | MSE (noise prediction) | | |

**Training Procedure for DCFG.**  The training of our proposed DCFG follows the same setup as standard Classifier-Free Guidance (CFG). Specifically, we apply classifier-free dropout (Ho & Salimans, 2022) by replacing the entire conditioning vector with a null token (i.e., zero vector) with probability $p_\varnothing = 0.5$. Unlike the typical choice of $p_\varnothing = 0.2$, we found that using $p_\varnothing = 0.5$ better preserves identity, which is particularly important for counterfactual generation tasks. Note that we apply dropout to all attributes jointly, rather than selectively masking subsets. One could alternatively consider group-wise dropout—nullifying only a random subset of attribute groups—but such partial masking may encourage the model to over-rely on the remaining visible attributes, making the resulting guidance less disentangled and less robust. We leave this as an interesting direction for future exploration.

**Computation Resources.**  All experiments were conducted on servers equipped with multiple NVIDIA GPUs, including L40S and similar models, each with approximately 48GB of memory. Training each model typically takes around one week on one GPU. Due to the high computational cost of diffusion-based sampling, generating counterfactuals for each intervention (do(key)) and each guidance configuration takes approximately one day for the MIMIC-CXR and EMBED datasets, and around 7 hours for CelebA-HQ.

**Evaluation.**  Due to the computational cost of diffusion sampling, we evaluate on fixed, balanced subsets rather than the full test sets. For CelebA-HQ and EMBED, we use 1,000 samples each, selected to ensure an even distribution across the conditioning attributes. For MIMIC-CXR, we evaluate on 1,500 samples, stratified to balance race groups. These fixed subsets are reused across all experiments to enable fair and consistent comparisons. To generate counterfactuals, we use DDIM sampling with 1,000 time steps, as we find this setting achieves stronger identity preservation compared to shorter schedules—an essential property for counterfactual evaluation. This setup allows us to assess attribute-specific phenomena such as amplification and reversibility while keeping the sampling cost manageable.

# D  CELEBA-HQ

## D.1  DATASET DETAILS

For the CelebA-HQ dataset (Karras et al., 2017), we select `Smiling`, `Male`, and `Young` as three independent binary variables. These are among the most reliably annotated attributes in CelebA, each achieving over 95% consistency in manual labeling (Wu et al., 2023). Moreover, they exhibit low inconsistency across duplicate face pairs (e.g., `Male`: 0.005; `Smiling`: 0.077), suggesting minimal label noise. We assume these variables to be independent, as our goal is to isolate and analyze attribute amplification under global classifier-free guidance (CFG), which is more interpretable with uncorrelated factors. As shown in Fig. A.1, the Pearson correlation matrix confirms weak pairwise correlations among these attributes. Although a moderate negative correlation is observed between `Male` and `Young` ($\rho = -0.33$), we attribute this to dataset bias rather than a true causal dependency, and proceed by modeling them as independent.

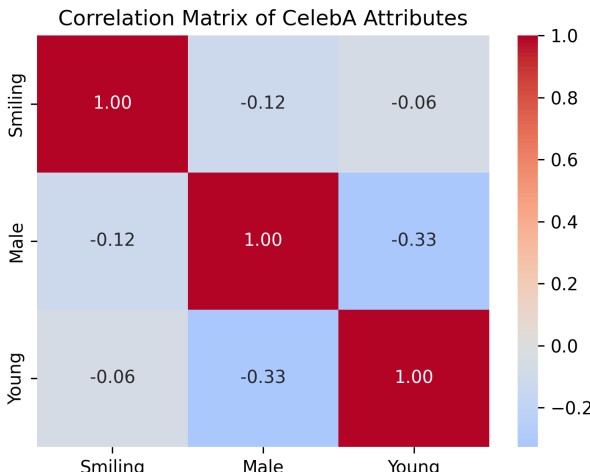

Figure A.1: Pearson correlation matrix of CelebA-HQ attributes: `Smiling`, `Male`, and `Young`. While a moderate negative correlation is observed between `Male` and `Young` ($\rho = -0.33$), we regard this as a spurious correlation likely stemming from dataset bias rather than a meaningful causal relationship. Therefore, for the purposes of our analysis, we assume these attributes to be independent.

### D.2 EXTRA QUANTITATIVE RESULT FOR CELEBA-HQ

Table A.3: CelebA-HQ: Effectiveness (ROC-AUC ↑) and Reversibility (MAE, LPIPS ↓) metrics when changing $\omega_{\text{aff}}$. Compared to global CFG (i.e., $\omega$), DCFG achieves strong intervention effectiveness on the intervened variable while mitigating amplification on invariant variables. For higher $\omega_{\text{aff}}$, we apply $\omega_{\text{inv}}=1.2$ to prevent degradation of invariant attributes. While reversibility deteriorates with increasing $\omega_{\text{aff}}$, DCFG consistently maintains better reversibility than global CFG with $\omega=\omega_{\text{aff}}$.

| do(key) | Guidance configuration | Smiling AUC/Δ | Male AUC/Δ | Young AUC/Δ | MAE | LPIPS |
|---|---|---|---|---|---|---|
| | $\omega=1.0$ | 86.5 / +0.0 | 96.9 / +0.0 | 78.6 / +0.0 | 0.113 | 0.082 |
| | $\omega=1.2$ | 91.7 / +5.2 | 98.5 / +1.6 | 80.9 / +2.3 | 0.133 | 0.091 |
| | $\omega=1.5$ | 95.5 / +9.0 | 99.2 / +2.3 | 84.3 / +5.7 | 0.163 | 0.111 |
| | $\omega=1.7$ | 96.7 / +10.2 | 99.4 / +2.5 | 85.7 / +7.1 | 0.179 | 0.119 |
| | $\omega=2.0$ | 97.7 / +11.2 | 99.6 / +2.7 | 88.3 / +9.7 | 0.203 | 0.127 |
| | $\omega=2.5$ | 98.6 / +12.1 | 99.7 / +2.8 | 89.3 / +10.7 | 0.234 | 0.142 |
| do(Smiling) | $\omega=3.0$ | 98.8 / +12.3 | 99.9 / +3.0 | 90.3 / +11.7 | 0.263 | 0.155 |
| | $\omega_{\text{aff}}=1.2, \omega_{\text{inv}}=1.0$ | 89.9 / +3.4 | 95.8 / -1.1 | 77.8 / -0.8 | 0.128 | 0.093 |
| | $\omega_{\text{aff}}=1.5, \omega_{\text{inv}}=1.2$ | 93.9 / +7.4 | 97.5 / +0.6 | 79.7 / +1.1 | 0.136 | 0.092 |
| | $\omega_{\text{aff}}=1.7, \omega_{\text{inv}}=1.2$ | 95.4 / +8.9 | 97.5 / +0.6 | 79.9 / +1.3 | 0.141 | 0.095 |
| | $\omega_{\text{aff}}=2.0, \omega_{\text{inv}}=1.2$ | 97.0 / +10.5 | 97.3 / +0.4 | 79.7 / +1.1 | 0.146 | 0.098 |
| | $\omega_{\text{aff}}=2.5, \omega_{\text{inv}}=1.2$ | 98.9 / +12.4 | 96.1 / -0.8 | 77.8 / -0.8 | 0.164 | 0.112 |
| | $\omega_{\text{aff}}=3.0, \omega_{\text{inv}}=1.2$ | 99.6 / +13.1 | 95.4 / -1.5 | 77.6 / -1.0 | 0.177 | 0.122 |
| | $\omega=1.0$ | 86.6 / +0.0 | 91.8 / +0.0 | 79.8 / +0.0 | 0.115 | 0.079 |
| | $\omega=1.2$ | 90.1 / +3.5 | 95.1 / +3.3 | 80.8 / +1.0 | 0.127 | 0.088 |
| | $\omega=1.5$ | 93.3 / +6.7 | 97.2 / +5.4 | 82.0 / +2.2 | 0.158 | 0.111 |
| | $\omega=1.7$ | 94.7 / +8.1 | 97.5 / +5.7 | 83.7 / +3.9 | 0.175 | 0.123 |
| | $\omega=2.0$ | 96.0 / +9.4 | 97.9 / +6.1 | 85.0 / +5.2 | 0.202 | 0.139 |
| | $\omega=2.5$ | 97.6 / +11.0 | 98.4 / +6.6 | 87.5 / +7.7 | 0.238 | 0.156 |
| do(Male) | $\omega=3.0$ | 98.2 / +11.6 | 99.2 / +7.4 | 90.2 / +10.4 | 0.267 | 0.171 |
| | $\omega_{\text{aff}}=1.2, \omega_{\text{inv}}=1.0$ | 85.1 / -1.5 | 91.3 / -0.5 | 79.0 / -0.8 | 0.137 | 0.097 |
| | $\omega_{\text{aff}}=1.5, \omega_{\text{inv}}=1.2$ | 88.3 / +1.7 | 93.8 / +2.0 | 78.9 / -0.9 | 0.149 | 0.101 |
| | $\omega_{\text{aff}}=1.7, \omega_{\text{inv}}=1.2$ | 88.1 / +1.5 | 95.8 / +4.0 | 77.5 / -2.3 | 0.151 | 0.103 |
| | $\omega_{\text{aff}}=2.0, \omega_{\text{inv}}=1.2$ | 88.0 / +1.4 | 97.8 / +6.0 | 77.0 / -2.8 | 0.158 | 0.109 |
| | $\omega_{\text{aff}}=2.5, \omega_{\text{inv}}=1.2$ | 87.4 / +0.8 | 99.4 / +7.6 | 76.2 / -3.6 | 0.171 | 0.118 |
| | $\omega_{\text{aff}}=3.0, \omega_{\text{inv}}=1.2$ | 87.4 / +0.8 | 99.7 / +7.9 | 75.9 / -3.9 | 0.188 | 0.130 |
| | $\omega=1.0$ | 87.5 / +0.0 | 95.7 / +0.0 | 62.3 / +0.0 | 0.115 | 0.085 |
| | $\omega=1.2$ | 90.8 / +3.3 | 97.4 / +1.7 | 64.5 / +2.2 | 0.130 | 0.088 |
| | $\omega=1.5$ | 95.6 / +8.1 | 99.3 / +3.6 | 66.3 / +4.0 | 0.166 | 0.110 |
| | $\omega=1.7$ | 96.7 / +9.2 | 99.4 / +3.7 | 67.8 / +5.5 | 0.183 | 0.119 |
| | $\omega=2.0$ | 97.7 / +10.2 | 99.6 / +3.9 | 69.5 / +7.2 | 0.204 | 0.130 |
| | $\omega=2.5$ | 98.3 / +10.8 | 99.8 / +4.1 | 73.5 / +11.2 | 0.234 | 0.146 |
| do(Young) | $\omega=3.0$ | 98.5 / +11.0 | 99.9 / +4.2 | 77.7 / +15.4 | 0.261 | 0.160 |
| | $\omega_{\text{aff}}=1.2, \omega_{\text{inv}}=1.0$ | 87.4 / -0.1 | 95.0 / -0.7 | 63.2 / +0.9 | 0.129 | 0.095 |
| | $\omega_{\text{aff}}=1.5, \omega_{\text{inv}}=1.2$ | 90.0 / +2.5 | 96.7 / +1.0 | 67.4 / +5.1 | 0.147 | 0.100 |
| | $\omega_{\text{aff}}=1.7, \omega_{\text{inv}}=1.2$ | 89.2 / +1.7 | 96.1 / +0.4 | 71.3 / +9.0 | 0.150 | 0.103 |
| | $\omega_{\text{aff}}=2.0, \omega_{\text{inv}}=1.2$ | 87.9 / +0.4 | 94.5 / -1.2 | 75.6 / +13.3 | 0.157 | 0.110 |
| | $\omega_{\text{aff}}=2.5, \omega_{\text{inv}}=1.2$ | 88.5 / +1.0 | 91.9 / -3.8 | 81.8 / +19.5 | 0.172 | 0.125 |
| | $\omega_{\text{aff}}=3.0, \omega_{\text{inv}}=1.2$ | 86.7 / -0.8 | 90.0 / -5.7 | 87.6 / +25.3 | 0.188 | 0.136 |

Table A.4: CelebA-HQ: Effectiveness (ROC-AUC ↑) and Reversibility (MAE, LPIPS ↓) metrics when changing $\omega_{\text{inv}}$. Increasing $\omega_{\text{inv}}$ consistently increases effectiveness on invariant variables, while degrading intervention effectiveness. When $\omega_{\text{inv}}=2.5$, the amplification on invariant attributes becomes comparable to that of the global CFG setting with $\omega=2.5$.

| do(key) | Guidance configuration | Smiling AUC/Δ | Male AUC/Δ | Young AUC/Δ | MAE | LPIPS |
|---|---|---|---|---|---|---|
| | $\omega=1.0$ | 86.5 / +0.0 | 96.9 / +0.0 | 78.6 / +0.0 | 0.113 | 0.082 |
| | $\omega=2.5$ | 98.6 / +12.1 | 99.7 / +2.8 | 89.3 / +10.7 | 0.234 | 0.142 |
| | $\omega_{\text{aff}}=2.5, \omega_{\text{inv}}=1.0$ | 99.1 / +12.6 | 92.6 / -4.3 | 75.2 / -3.4 | 0.165 | 0.118 |
| do(Smiling) | $\omega_{\text{aff}}=2.5, \omega_{\text{inv}}=1.2$ | 98.9 / +12.4 | 96.1 / -0.8 | 77.8 / -0.8 | 0.164 | 0.112 |
| | $\omega_{\text{aff}}=2.5, \omega_{\text{inv}}=1.5$ | 98.3 / +11.8 | 98.2 / +1.3 | 82.6 / +4.0 | 0.177 | 0.118 |
| | $\omega_{\text{aff}}=2.5, \omega_{\text{inv}}=1.7$ | 98.1 / +11.6 | 98.8 / +1.9 | 84.0 / +5.4 | 0.189 | 0.123 |
| | $\omega_{\text{aff}}=2.5, \omega_{\text{inv}}=2.0$ | 97.5 / +11.0 | 99.3 / +2.4 | 87.0 / +8.4 | 0.209 | 0.131 |
| | $\omega_{\text{aff}}=2.5, \omega_{\text{inv}}=2.5$ | 96.3 / +9.8 | 99.5 / +2.6 | 88.7 / +10.1 | 0.236 | 0.143 |
| | $\omega=1.0$ | 86.6 / +0.0 | 91.8 / +0.0 | 79.8 / +0.0 | 0.115 | 0.079 |
| | $\omega=2.5$ | 97.6 / +11.0 | 98.4 / +6.6 | 87.5 / +7.7 | 0.238 | 0.156 |
| | $\omega_{\text{aff}}=2.5, \omega_{\text{inv}}=1.0$ | 83.4 / -3.2 | 99.4 / +7.6 | 68.9 / -10.9 | 0.173 | 0.122 |
| do(Male) | $\omega_{\text{aff}}=2.5, \omega_{\text{inv}}=1.2$ | 87.4 / +0.8 | 99.4 / +7.6 | 71.1 / -8.7 | 0.171 | 0.118 |
| | $\omega_{\text{aff}}=2.5, \omega_{\text{inv}}=1.5$ | 92.0 / +5.4 | 99.3 / +7.5 | 74.0 / -5.8 | 0.182 | 0.119 |
| | $\omega_{\text{aff}}=2.5, \omega_{\text{inv}}=1.7$ | 93.5 / +6.9 | 98.9 / +7.1 | 76.8 / -3.0 | 0.189 | 0.123 |
| | $\omega_{\text{aff}}=2.5, \omega_{\text{inv}}=2.0$ | 95.3 / +8.7 | 98.7 / +6.9 | 80.2 / +0.4 | 0.207 | 0.135 |
| | $\omega_{\text{aff}}=2.5, \omega_{\text{inv}}=2.5$ | 97.2 / +10.6 | 97.7 / +5.9 | 87.5 / +7.7 | 0.242 | 0.158 |
| | $\omega=1.0$ | 87.5 / +0.0 | 95.7 / +0.0 | 62.3 / +0.0 | 0.115 | 0.085 |
| | $\omega=2.5$ | 98.3 / +10.8 | 99.8 / +4.1 | 73.5 / +11.2 | 0.234 | 0.146 |
| | $\omega_{\text{aff}}=2.5, \omega_{\text{inv}}=1.0$ | 83.4 / -4.1 | 85.9 / -9.8 | 85.1 / +22.8 | 0.169 | 0.127 |
| do(Young) | $\omega_{\text{aff}}=2.5, \omega_{\text{inv}}=1.2$ | 88.5 / +1.0 | 91.9 / -3.8 | 81.8 / +19.5 | 0.172 | 0.125 |
| | $\omega_{\text{aff}}=2.5, \omega_{\text{inv}}=1.5$ | 92.4 / +4.9 | 96.4 / +0.7 | 78.5 / +16.2 | 0.187 | 0.127 |
| | $\omega_{\text{aff}}=2.5, \omega_{\text{inv}}=1.7$ | 94.3 / +6.8 | 97.8 / +2.1 | 75.3 / +13.0 | 0.199 | 0.133 |
| | $\omega_{\text{aff}}=2.5, \omega_{\text{inv}}=2.0$ | 97.1 / +9.6 | 99.0 / +3.3 | 73.3 / +11.0 | 0.215 | 0.139 |
| | $\omega_{\text{aff}}=2.5, \omega_{\text{inv}}=2.5$ | 99.3 / +11.8 | 99.7 / +4.0 | 68.5 / +6.2 | 0.238 | 0.147 |

## D.3 EXTRA QUALITATIVE RESULTS FOR CELEBA-HQ

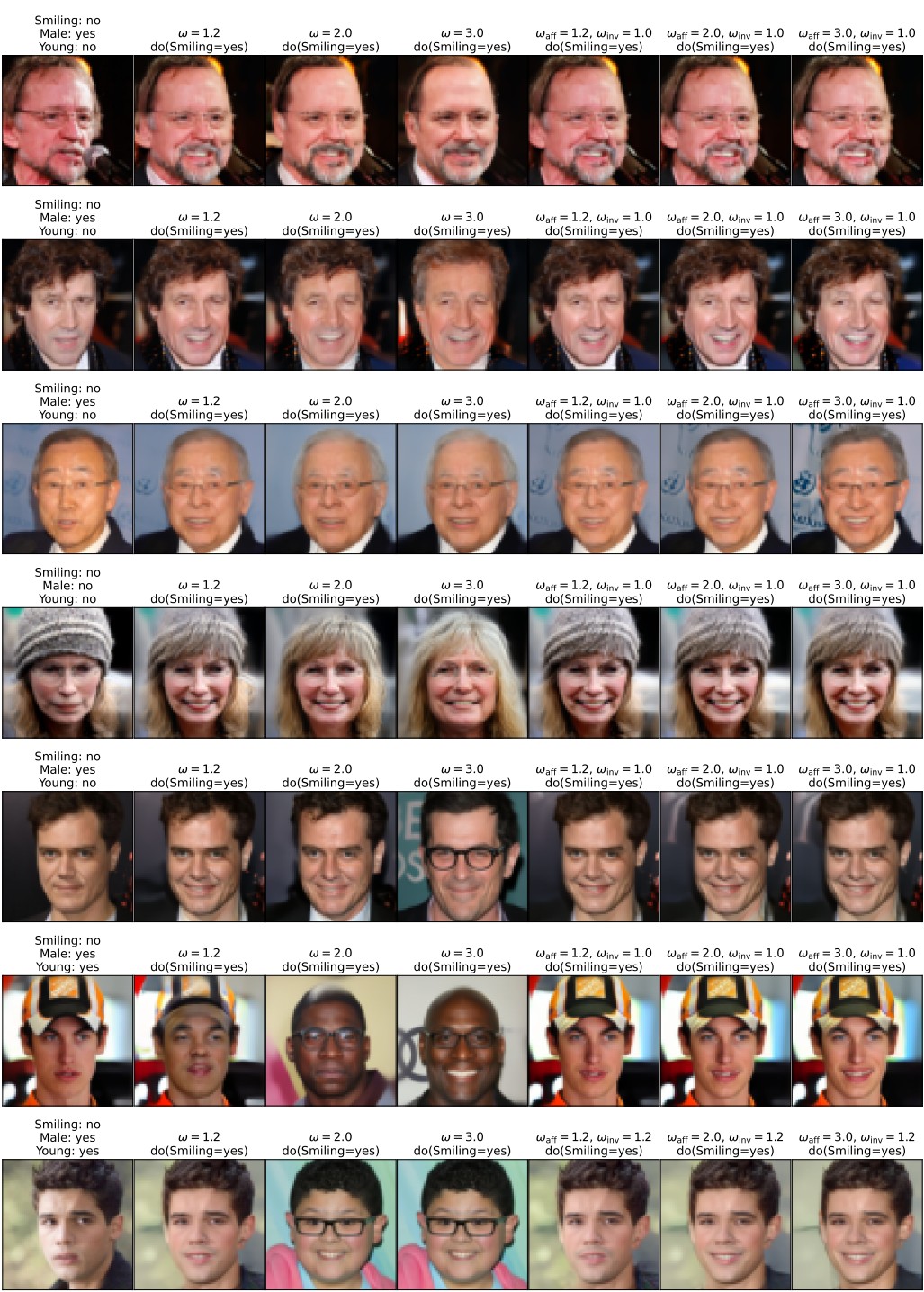

Figure A.2: **Additional qualitative results for do(Smiling) on CelebA-HQ.** Each row shows the original image followed by counterfactuals generated with global CFG ($\omega$) and DCFG ($\omega_{\text{aff}}, \omega_{\text{inv}}$). DCFG better preserves *invariant* attributes and identity while effectively reflecting the intervention.

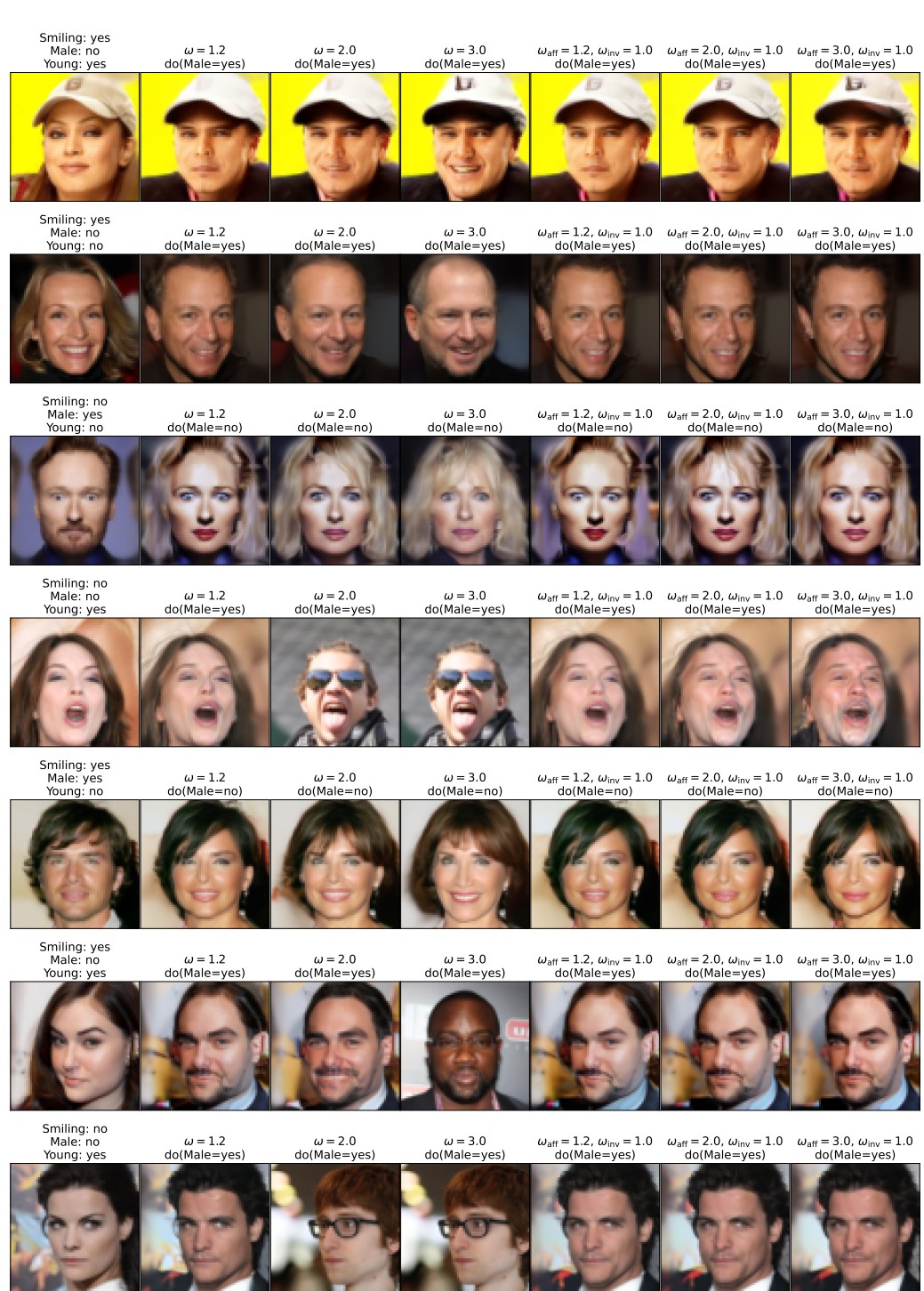

Figure A.3: **Additional qualitative results for `do(Male)` on CelebA-HQ.** Each row shows the original image followed by counterfactuals generated with global CFG ($\omega$) and DCFG ($\omega_{\text{aff}}, \omega_{\text{inv}}$). DCFG better preserves *invariant* attributes and identity while effectively reflecting the intervention.

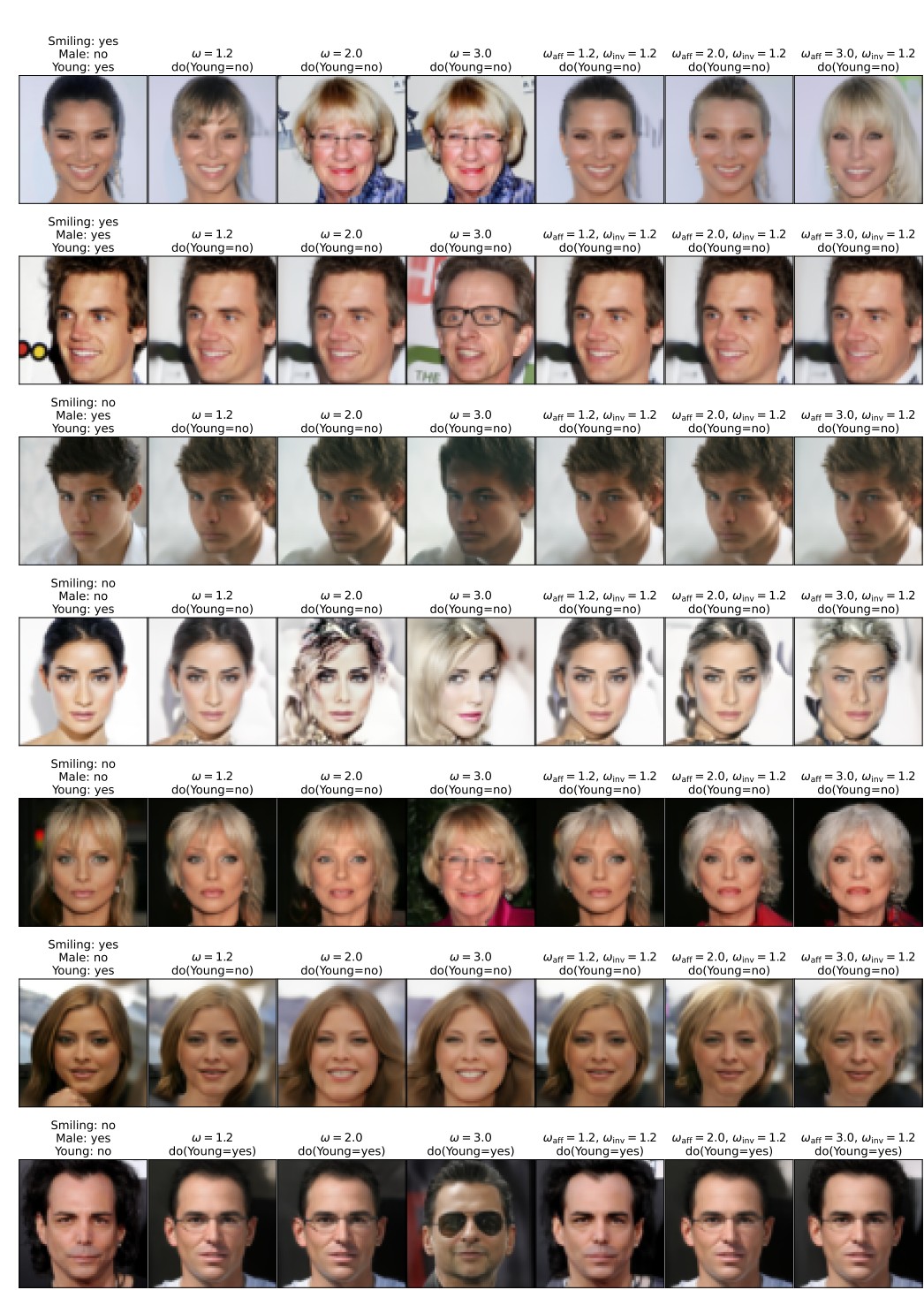

Figure A.4: **Additional qualitative results for do(Young) on CelebA-HQ.** Each row shows the original image followed by counterfactuals generated with global CFG ($\omega$) and DCFG ($\omega_{\text{aff}}$, $\omega_{\text{inv}}$). DCFG better preserves *invariant* attributes and identity while effectively reflecting the intervention.

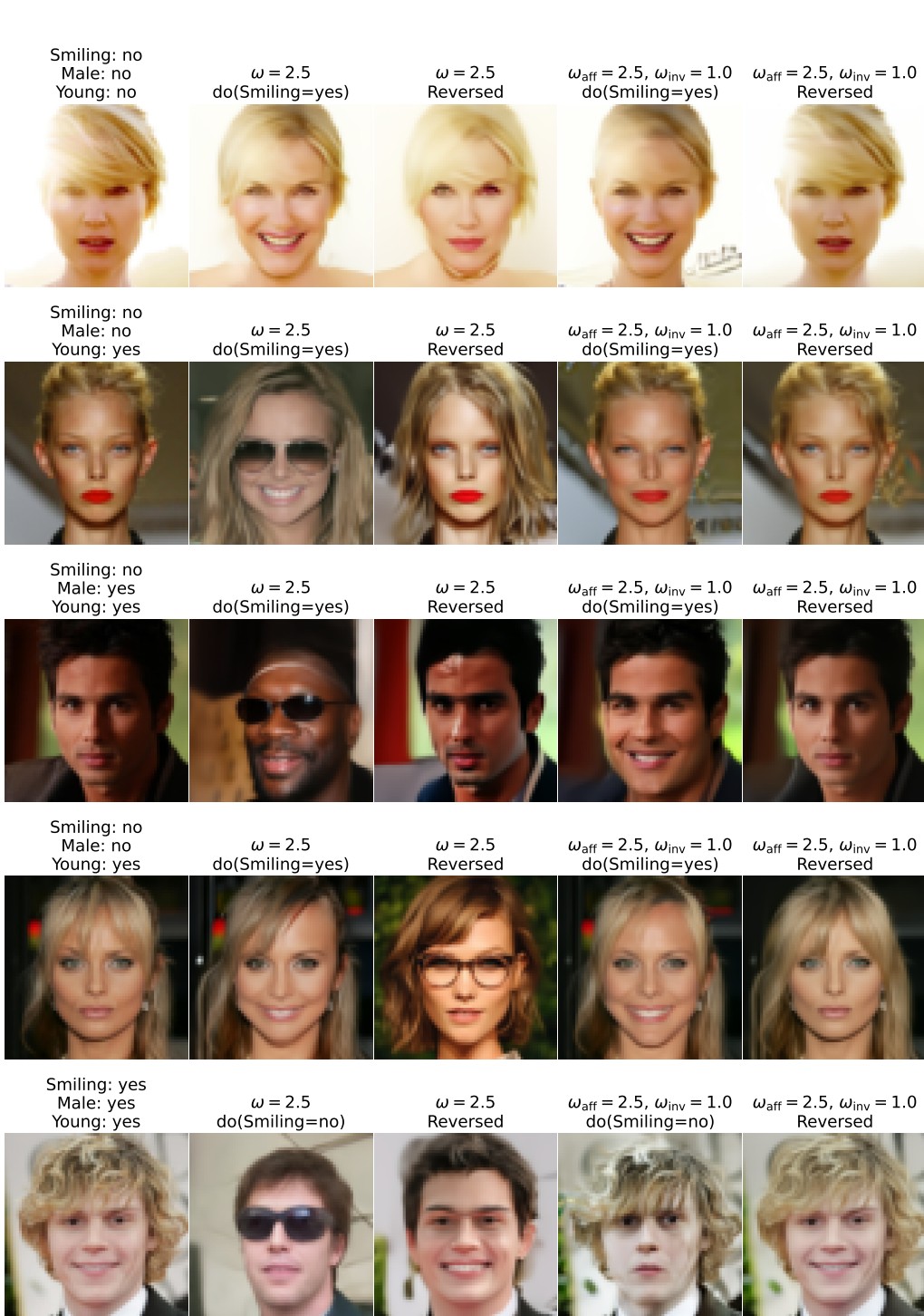

Figure A.5: **Reversibility analysis for `do(Similing)` on CelebA-HQ.** Each row shows the original image, followed by counterfactuals generated using global CFG ($\omega$) and our proposed DCFG ($\omega_{int}, \omega_{inv}$), along with their respective reversed generations. DCFG more faithfully preserves non-intervened attributes, resulting in visually and semantically more consistent reversals. This highlights the benefit of DCFG in enhancing both targeted editability and reversibility.

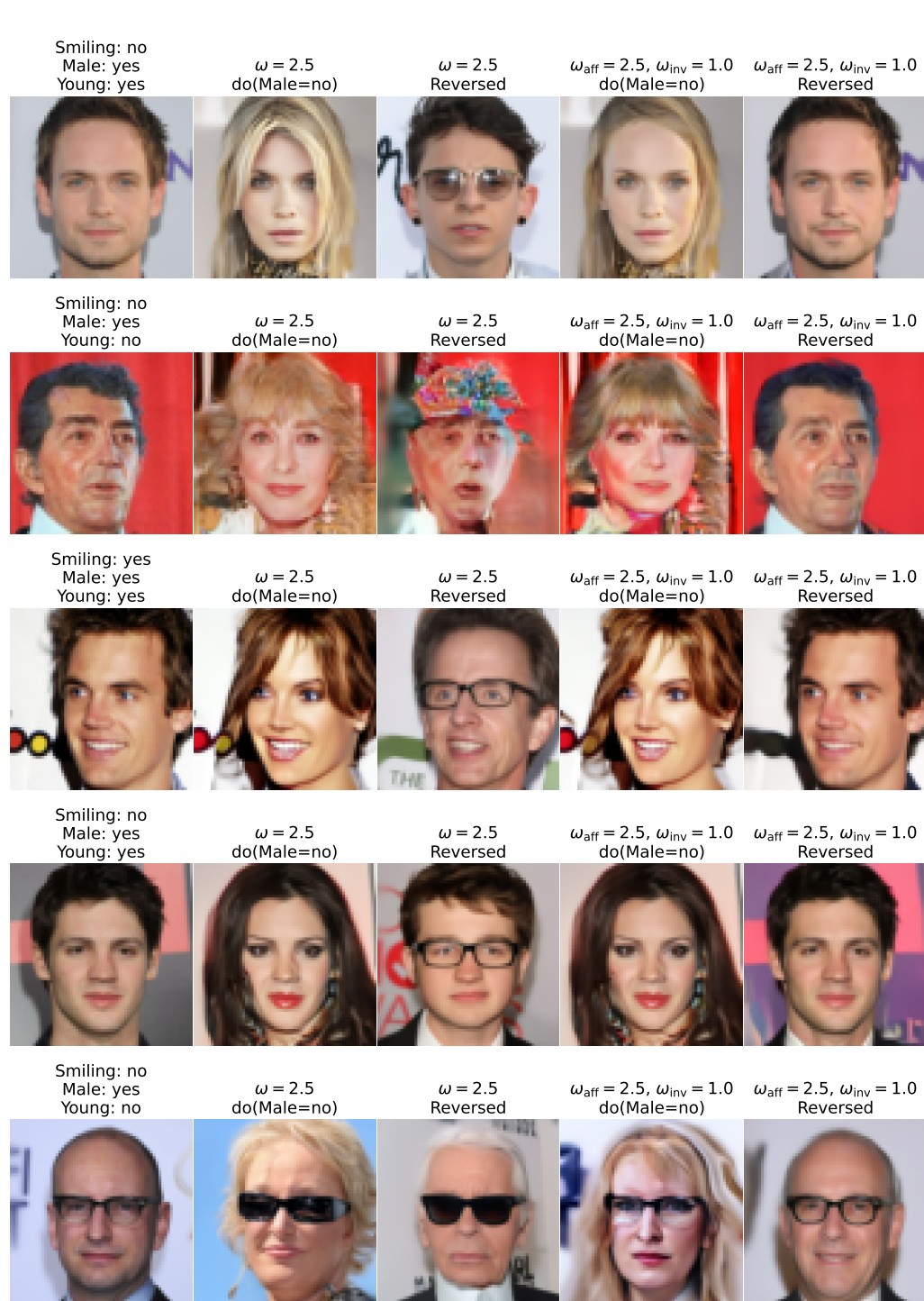

Figure A.6: **Reversibility analysis for do(Male) on CelebA-HQ.** Each row shows the original image, followed by counterfactuals generated using global CFG ($\omega$) and our proposed DCFG ($\omega_{int}, \omega_{inv}$), along with their respective reversed generations. DCFG more faithfully preserves non-intervened attributes, resulting in visually and semantically more consistent reversals. This highlights the benefit of DCFG in enhancing both targeted editability and reversibility.

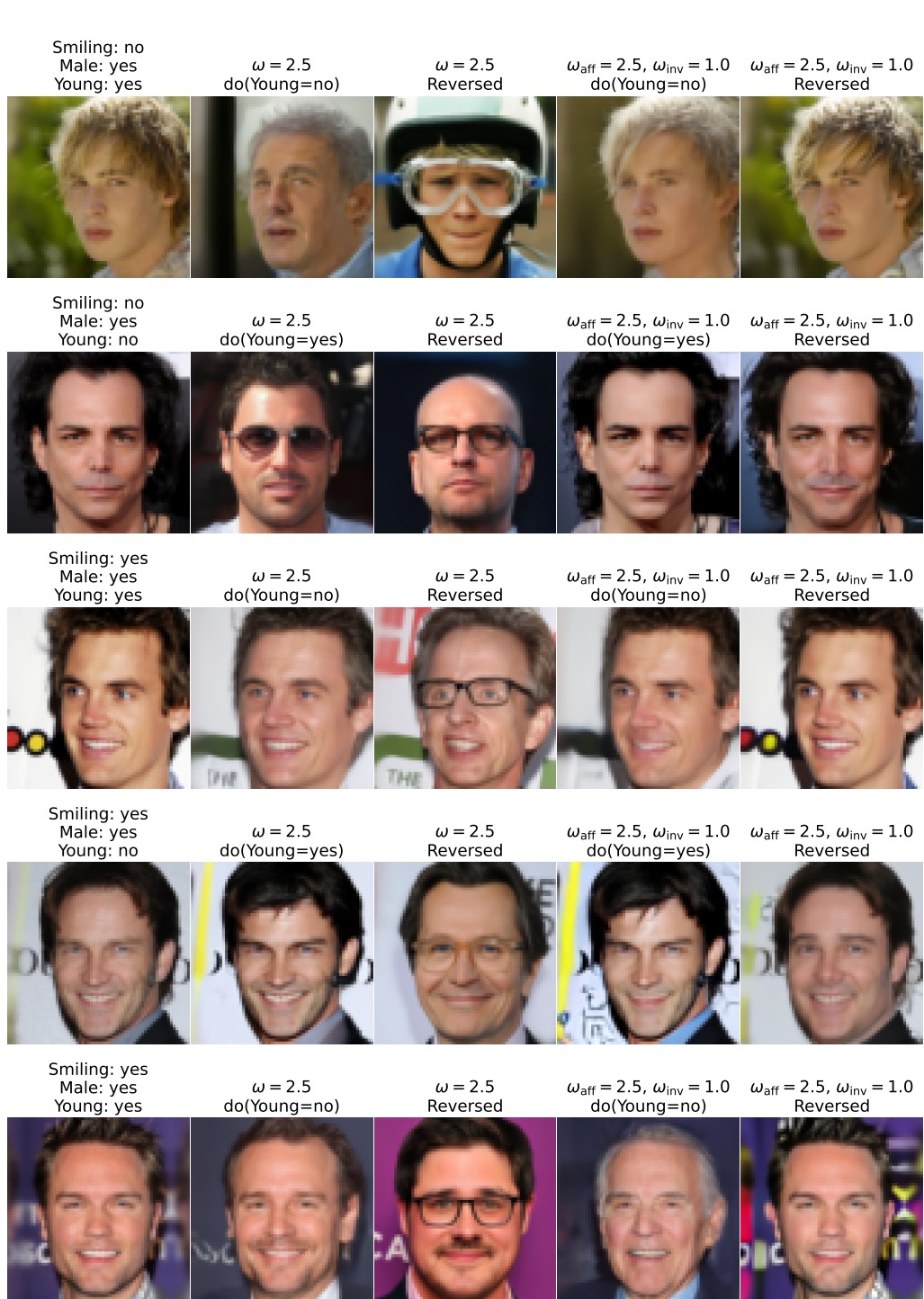

Figure A.7: **Reversibility analysis for do(Young) on CelebA-HQ.** Each row shows the original image, followed by counterfactuals generated using global CFG ($\omega$) and our proposed DCFG ($\omega_{int}, \omega_{inv}$), along with their respective reversed generations. DCFG more faithfully preserves non-intervened attributes, resulting in visually and semantically more consistent reversals. This highlights the benefit of DCFG in enhancing both targeted editability and reversibility.

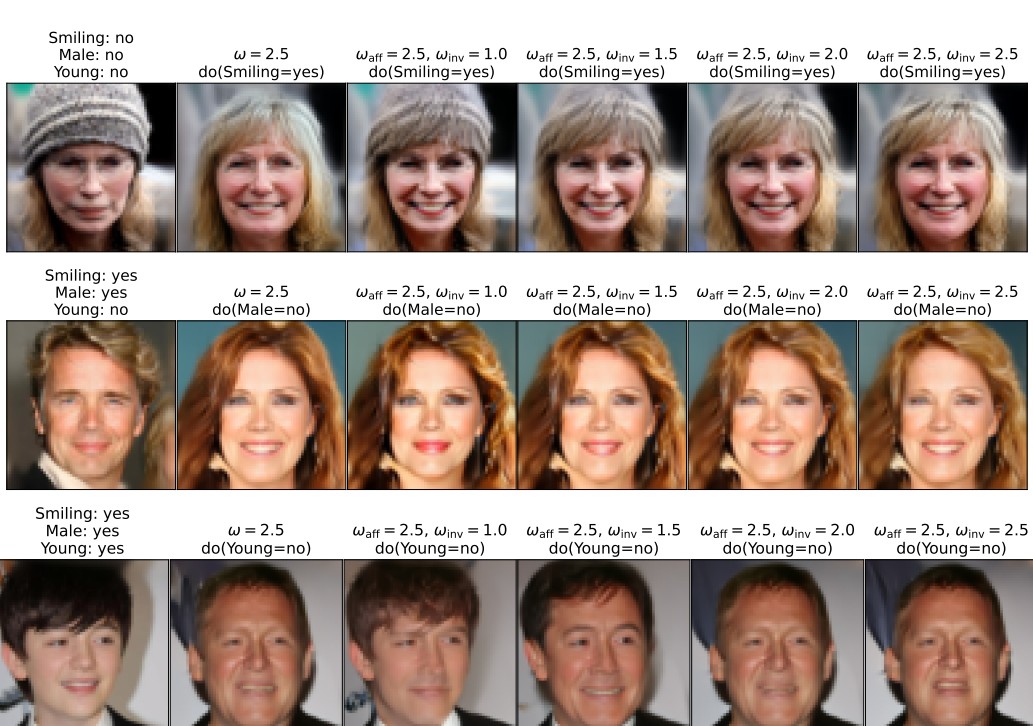

Figure A.8: **Effect of $\omega_{\text{inv}}$ on CelebA-HQ counterfactuals.** Each row shows the original image followed by counterfactuals generated using global CFG ($\omega$=2.5) and our proposed DCFG with fixed intervention guidance ($\omega_{\text{aff}}$=2.5) and varying invariant guidance $\omega_{\text{inv}} \in \{1.0, 1.5, 2.0, 2.5\}$. As $\omega_{\text{inv}}$ increases, amplification of invariant attributes becomes more pronounced, and at $\omega_{\text{inv}}$=2.5, DCFG effectively reproduces the same over-editing behavior as global CFG. This shows that $\omega_{\text{inv}}$ modulates the degree of guidance applied to invariant attributes and should be carefully calibrated to maintain identity and disentanglement.

# E EMEBD

## E.1 DATASET DETAILS

We use the EMory BrEast imaging Dataset (EMBED) (Jeong et al., 2023) for our experiments. Schueppert et al. (2024) manually labeled 22,012 images with circular markers and trained a classifier on this subset, which was then applied to the full dataset to infer circle annotations. We adopt this preprocessing pipeline and extract the `circle` attribute from their predictions. To define the `density` label, we binarize the original four-category breast density annotations by grouping categories A and B as `low` density, and categories C and D as `high` density. While the full dataset comprises 151,948 training, 7,156 validation, and 43,669 test samples, we use only 1,000 test samples in this work due to the high computational cost of diffusion models. As shown in Fig. A.9, the Pearson correlation matrix reveals that `density` and `circle` are nearly uncorrelated, supporting our assumption of their independence.

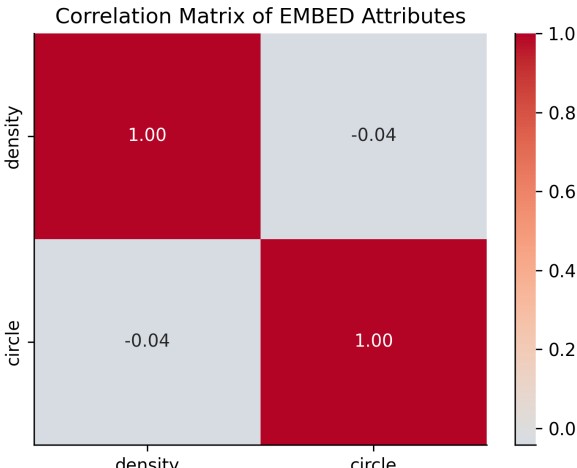

Figure A.9: **Pearson correlation matrix of EMBED attributes:** `density` and `circle`. The correlation between these two variables is negligible ($\rho = -0.04$), suggesting that they can be reasonably treated as independent for the purposes of our analysis.

## E.2 EXTRA QUANTITATIVE RESULTS FOR EMBED

Table A.5: EMBED: Effectiveness (ROC-AUC ↑) and Reversibility (MAE, LPIPS ↓) metrics when changing $\omega_{\text{aff}}$. Compared to global CFG (i.e., $\omega$), DCFG achieves strong intervention effectiveness on the intervened variable while mitigating amplification on invariant variables. For higher $\omega_{\text{aff}}$, we apply $\omega_{\text{inv}}{=}1.2$ to prevent degradation of invariant attributes. While reversibility deteriorates with increasing $\omega_{\text{aff}}$, DCFG consistently maintains better reversibility than global CFG with $\omega{=}\omega_{\text{aff}}$.

| do(key) | Guidance configuration | Density AUC/$\Delta$ | Circle AUC/$\Delta$ | MAE | LPIPS |
|---|---|---|---|---|---|
| do(density) | $\omega{=}1.0$ | 63.4 / +0.0 | 92.9 / +0.0 | 0.027 | 0.033 |
| | $\omega{=}1.2$ | 70.5 / +7.1 | 94.5 / +1.6 | 0.033 | 0.038 |
| | $\omega{=}1.5$ | 79.0 / +15.6 | 95.9 / +3.0 | 0.035 | 0.047 |
| | $\omega{=}1.7$ | 84.3 / +20.9 | 96.7 / +3.8 | 0.032 | 0.055 |
| | $\omega{=}2.0$ | 89.6 / +26.2 | 97.5 / +4.6 | 0.034 | 0.064 |
| | $\omega{=}2.5$ | 95.2 / +31.8 | 97.7 / +4.8 | 0.042 | 0.076 |
| | $\omega{=}3.0$ | 97.8 / +34.4 | 98.2 / +5.3 | 0.045 | 0.086 |
| | $\omega_{\text{aff}}{=}1.2, \omega_{\text{inv}}{=}1.0$ | 73.1 / +9.7 | 92.8 / -0.1 | 0.028 | 0.038 |
| | $\omega_{\text{aff}}{=}1.5, \omega_{\text{inv}}{=}1.0$ | 81.6 / +18.2 | 92.2 / -0.7 | 0.029 | 0.043 |
| | $\omega_{\text{aff}}{=}1.7, \omega_{\text{inv}}{=}1.0$ | 86.2 / +22.8 | 91.6 / -1.3 | 0.031 | 0.048 |
| | $\omega_{\text{aff}}{=}2.0, \omega_{\text{inv}}{=}1.0$ | 91.6 / +28.2 | 90.7 / -2.2 | 0.032 | 0.053 |
| | $\omega_{\text{aff}}{=}2.5, \omega_{\text{inv}}{=}1.2$ | 96.6 / +33.2 | 92.2 / -0.7 | 0.036 | 0.064 |
| | $\omega_{\text{aff}}{=}3.0, \omega_{\text{inv}}{=}1.2$ | 98.6 / +35.2 | 91.6 / -1.3 | 0.038 | 0.071 |
| do(circle) | $\omega{=}1.0$ | 92.6 / +0.0 | 90.6 / +0.0 | 0.023 | 0.026 |
| | $\omega{=}1.2$ | 94.7 / +2.1 | 92.1 / +1.5 | 0.029 | 0.024 |
| | $\omega{=}1.5$ | 96.8 / +4.2 | 93.9 / +3.3 | 0.030 | 0.027 |
| | $\omega{=}1.7$ | 97.9 / +5.3 | 95.2 / +4.6 | 0.027 | 0.035 |
| | $\omega{=}2.0$ | 98.8 / +6.2 | 96.4 / +5.8 | 0.030 | 0.040 |
| | $\omega{=}2.5$ | 99.7 / +7.1 | 97.8 / +7.2 | 0.038 | 0.043 |
| | $\omega{=}3.0$ | 99.9 / +7.3 | 98.4 / +7.8 | 0.042 | 0.051 |
| | $\omega_{\text{aff}}{=}1.2, \omega_{\text{inv}}{=}1.0$ | 93.3 / +0.7 | 92.6 / +2.0 | 0.024 | 0.028 |
| | $\omega_{\text{aff}}{=}1.5, \omega_{\text{inv}}{=}1.0$ | 93.2 / +0.6 | 94.4 / +3.8 | 0.025 | 0.030 |
| | $\omega_{\text{aff}}{=}1.7, \omega_{\text{inv}}{=}1.0$ | 93.2 / +0.6 | 95.7 / +5.1 | 0.025 | 0.032 |
| | $\omega_{\text{aff}}{=}2.0, \omega_{\text{inv}}{=}1.0$ | 92.9 / +0.3 | 97.2 / +6.6 | 0.026 | 0.034 |
| | $\omega_{\text{aff}}{=}2.5, \omega_{\text{inv}}{=}1.2$ | 94.5 / +1.9 | 98.5 / +7.9 | 0.027 | 0.038 |
| | $\omega_{\text{aff}}{=}3.0, \omega_{\text{inv}}{=}1.2$ | 94.0 / +1.4 | 98.9 / +8.3 | 0.029 | 0.042 |

Table A.6: EMBED: Effectiveness (ROC-AUC ↑) and Reversibility (MAE, LPIPS ↓) metrics when changing $\omega_{\text{inv}}$. Increasing $\omega_{\text{inv}}$ consistently increases effectiveness on invariant variables, while degrading intervention effectiveness. When $\omega_{\text{inv}}{=}2.5$, the amplification on invariant attributes becomes comparable to that of the global CFG setting with $\omega{=}2.5$.

| do(key) | Guidance configuration | Density AUC/$\Delta$ | Circle AUC/$\Delta$ | MAE | LPIPS |
|---|---|---|---|---|---|
| do(density) | $\omega{=}1.0$ | 63.4 / +0.0 | 92.9 / +0.0 | 0.027 | 0.033 |
| | $\omega{=}2.5$ | 95.2 / +31.8 | 97.7 / +4.8 | 0.042 | 0.076 |
| | $\omega_{\text{aff}}{=}2.5, \omega_{\text{inv}}{=}1.0$ | 96.7 / +33.3 | 89.5 / -3.4 | 0.035 | 0.063 |
| | $\omega_{\text{aff}}{=}2.5, \omega_{\text{inv}}{=}1.2$ | 96.6 / +33.2 | 92.2 / -0.7 | 0.036 | 0.064 |
| | $\omega_{\text{aff}}{=}2.5, \omega_{\text{inv}}{=}1.5$ | 96.6 / +33.2 | 94.6 / +1.7 | 0.036 | 0.067 |
| | $\omega_{\text{aff}}{=}2.5, \omega_{\text{inv}}{=}1.7$ | 96.6 / +33.2 | 95.7 / +2.8 | 0.037 | 0.070 |
| | $\omega_{\text{aff}}{=}2.5, \omega_{\text{inv}}{=}2.0$ | 96.5 / +33.1 | 96.6 / +3.7 | 0.038 | 0.073 |
| | $\omega_{\text{aff}}{=}2.5, \omega_{\text{inv}}{=}2.5$ | 96.4 / +33.0 | 97.6 / +4.7 | 0.039 | 0.080 |
| do(circle) | $\omega{=}1.0$ | 92.6 / +0.0 | 90.6 / +0.0 | 0.023 | 0.026 |
| | $\omega{=}2.5$ | 99.7 / +7.1 | 97.8 / +7.2 | 0.038 | 0.043 |
| | $\omega_{\text{aff}}{=}2.5, \omega_{\text{inv}}{=}1.0$ | 92.2 / -0.4 | 98.5 / +7.9 | 0.028 | 0.039 |
| | $\omega_{\text{aff}}{=}2.5, \omega_{\text{inv}}{=}1.2$ | 94.5 / +1.9 | 98.5 / +7.9 | 0.027 | 0.038 |
| | $\omega_{\text{aff}}{=}2.5, \omega_{\text{inv}}{=}1.5$ | 97.2 / +4.6 | 98.3 / +7.7 | 0.028 | 0.039 |
| | $\omega_{\text{aff}}{=}2.5, \omega_{\text{inv}}{=}1.7$ | 98.1 / +5.5 | 98.2 / +7.6 | 0.029 | 0.040 |
| | $\omega_{\text{aff}}{=}2.5, \omega_{\text{inv}}{=}2.0$ | 99.0 / +6.4 | 98.2 / +7.6 | 0.031 | 0.043 |
| | $\omega_{\text{aff}}{=}2.5, \omega_{\text{inv}}{=}2.5$ | 99.8 / +7.2 | 98.0 / +7.4 | 0.035 | 0.050 |

### E.3 Extra qualitative results for EMBED

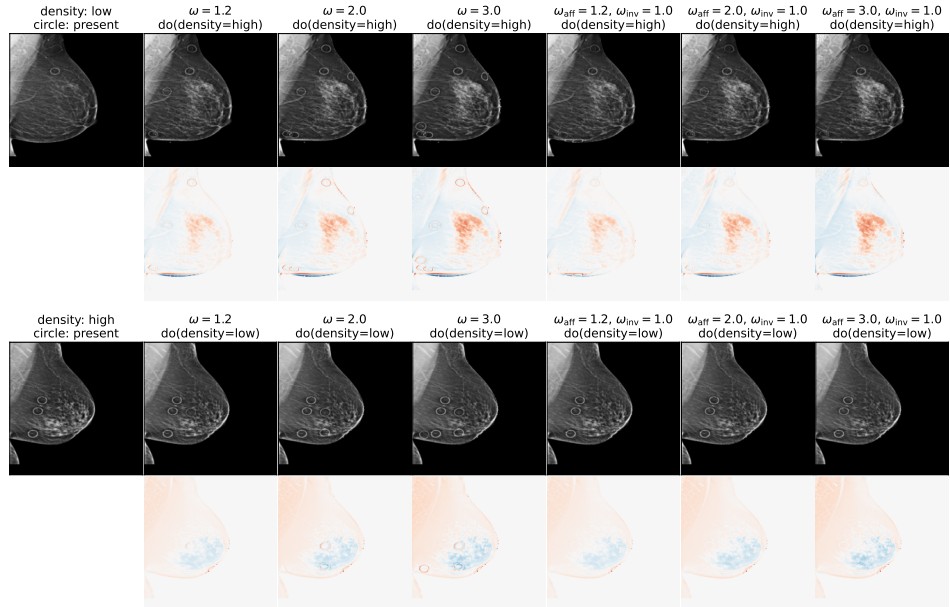

Figure A.10: **Additional qualitative results for do(density) on EMBED.** Each row shows the original image followed by counterfactuals generated with global CFG ($\omega$) and DCFG ($\omega_{\text{aff}}, \omega_{\text{inv}}$). DCFG better preserves *invariant* attributes and identity while effectively reflecting the intervention. Notably, under global CFG, increasing $\omega$ leads to spurious changes in circle count, whereas DCFG mitigates such amplification.

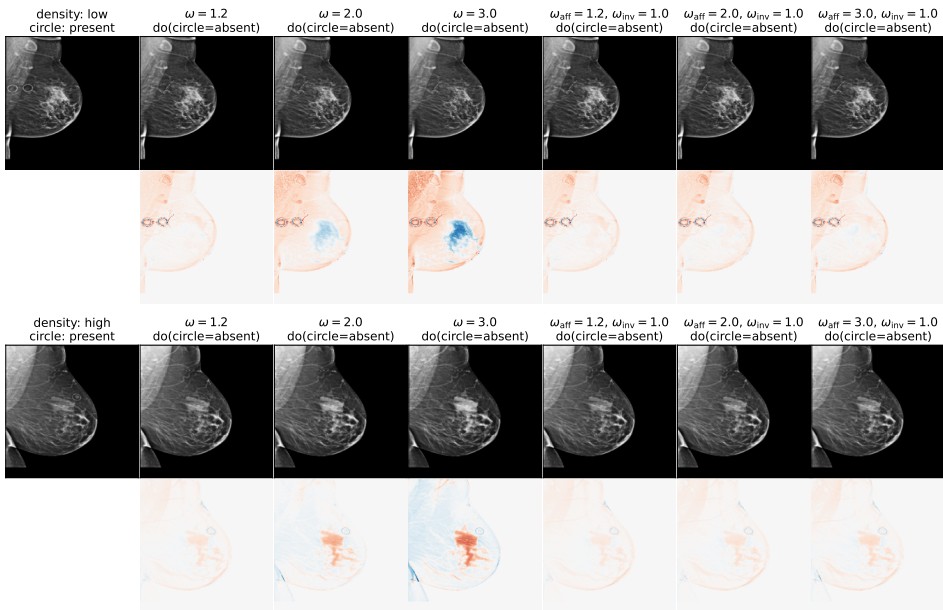

Figure A.11: **Additional qualitative results for do(circle) on EMBED.** Each row shows the original image followed by counterfactuals generated with global CFG ($\omega$) and DCFG ($\omega_{\text{aff}}, \omega_{\text{inv}}$) and the difference map (CF-input). DCFG better preserves *invariant* attributes and identity while effectively reflecting the intervention. Notably, under global CFG, increasing $\omega$ leads to spurious changes in density, whereas DCFG mitigates such amplification.

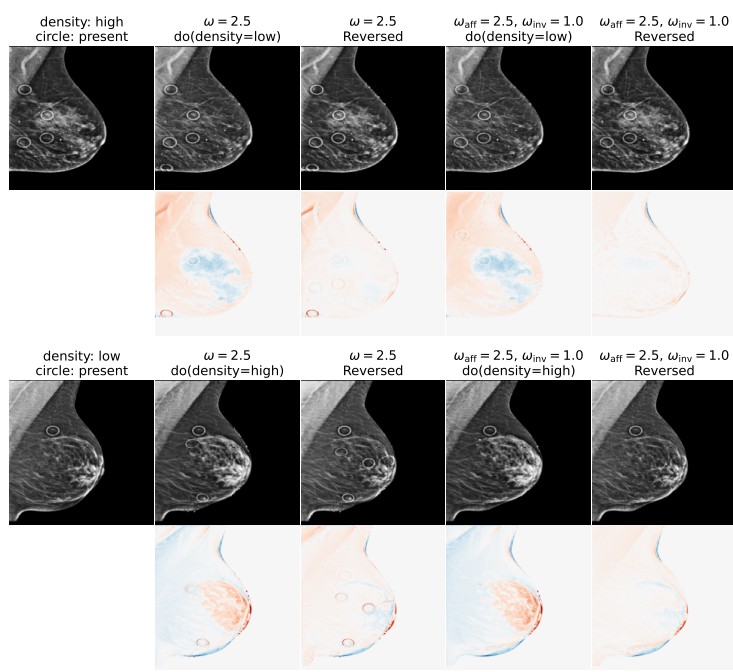

Figure A.12: **Reversibility analysis for `do(density)` on EMBED.** Each row shows the original image, the counterfactual generated using global CFG ($\omega$) or DCFG ($\omega_{int}, \omega_{inv}$), their corresponding reversed generations, and the associated difference maps (counterfactual - input, and reversed - input). DCFG more faithfully preserves non-intervened attributes and leads to smaller residuals in the difference maps, indicating better identity preservation.

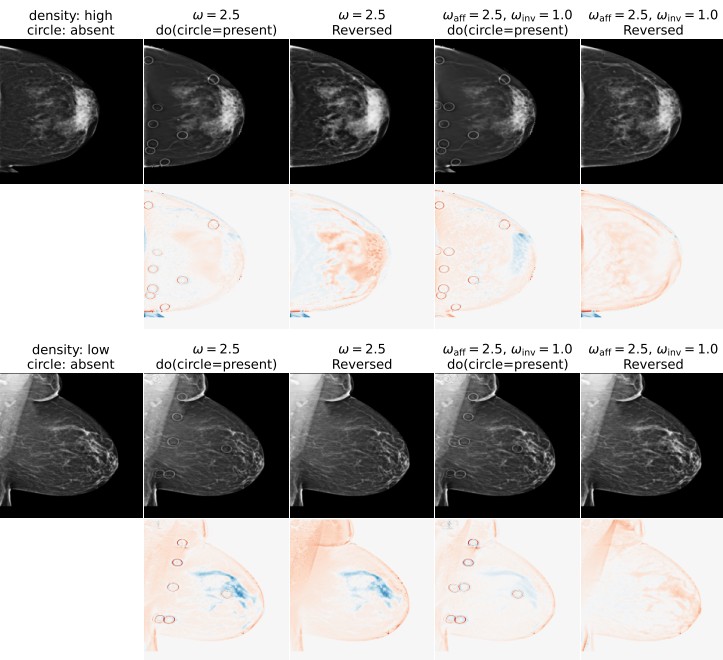

Figure A.13: **Reversibility analysis for `do(circle)` on EMBED.** Each row shows the original image, the counterfactual generated using global CFG ($\omega$) or DCFG ($\omega_{int}, \omega_{inv}$), their corresponding reversed generations, and the associated difference maps (counterfactual - input, and reversed - input). DCFG more faithfully preserves non-intervened attributes and leads to smaller residuals in the difference maps, indicating better identity preservation.

# F MIMIC

## F.1 DATASET DETAILS

We use the MIMIC-CXR dataset (Johnson et al., 2019) in our experiments. Following the dataset splits and filtering protocols of Ribeiro et al. (2023); Glocker et al. (2023), we focus on the binary disease label for pleural effusion. We adopt the same causal graph (DAG) as proposed in Ribeiro et al. (2023), in which age is modeled as a parent of finding. While we include age as part of the conditioning variables, we do not intervene on it. Instead, our primary goal is to study amplification of unintervened variables caused by CFG. For this purpose, we focus on race, sex, and finding, which we assume to be mutually independent. Fig. A.14 shows the Pearson correlation matrix of these three attributes, where all pairwise correlations are small (e.g., $\rho=0.12$ between race and sex, and $\rho=-0.15$ between race and finding), supporting the validity of the independence assumption in our counterfactual modeling.

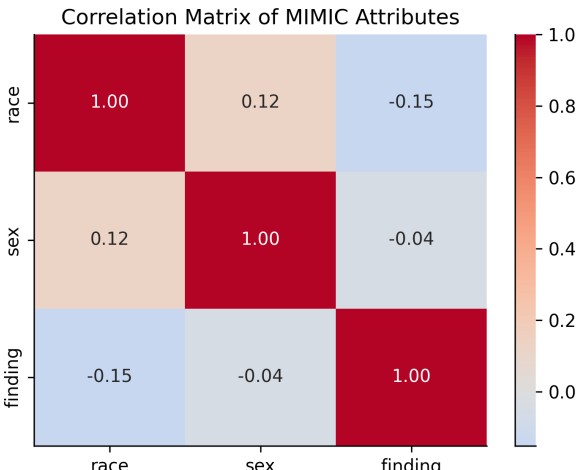

Figure A.14: **Pearson correlation matrix of MIMIC attributes:** race, sex, and finding. All pairwise correlations are low (e.g., $\rho=0.12$ between race and sex, and $\rho=-0.15$ between race and finding), suggesting that these variables can be reasonably treated as independent for the purposes of our counterfactual analysis.

## F.2 EXTRA QUANTITATIVE RESULTS FOR MIMIC-CXR

Table A.7: MIMIC: Effectiveness (ROC-AUC ↑) and Reversibility (MAE, LPIPS ↓) metrics when changing $\omega_{\text{aff}}$. Compared to global CFG (i.e., $\omega$), DCFG achieves strong intervention effectiveness on the intervened variable while mitigating amplification on invariant variables. For higher $\omega_{\text{aff}}$, we apply $\omega_{\text{inv}}{=}1.2$ to prevent degradation of invariant attributes. While reversibility tends to degrade as $\omega_{\text{aff}}$ increases, DCFG maintains better reversibility than global CFG at higher guidance strengths.

| do(key) | Guidance configuration | Sex AUC/Δ | Race AUC/Δ | Finding AUC/Δ | MAE | LPIPS |
|---|---|---|---|---|---|---|
| | $\omega{=}1.0$ | 92.4 / +0.0 | 75.6 / +0.0 | 88.8 / +0.0 | 0.146 | 0.202 |
| | $\omega{=}1.2$ | 95.2 / +2.8 | 79.3 / +3.7 | 92.8 / +4.0 | 0.151 | 0.206 |
| | $\omega{=}1.5$ | 97.7 / +5.3 | 82.4 / +6.8 | 95.7 / +6.9 | 0.171 | 0.226 |
| | $\omega{=}1.7$ | 98.5 / +6.1 | 84.7 / +9.1 | 97.0 / +8.2 | 0.186 | 0.239 |
| | $\omega{=}2.0$ | 99.3 / +6.9 | 87.4 / +11.8 | 98.0 / +9.2 | 0.207 | 0.258 |
| | $\omega{=}2.5$ | 99.8 / +7.4 | 90.5 / +14.9 | 99.0 / +10.2 | 0.239 | 0.284 |
| do(sex) | $\omega{=}3.0$ | 99.9 / +7.5 | 92.9 / +17.3 | 99.5 / +10.7 | 0.266 | 0.305 |
| | $\omega_{\text{aff}}{=}1.2, \omega_{\text{inv}}{=}1.0$ | 96.4 / +4.0 | 74.6 / -1.0 | 89.1 / +0.3 | 0.158 | 0.217 |
| | $\omega_{\text{aff}}{=}1.5, \omega_{\text{inv}}{=}1.0$ | 98.4 / +6.0 | 74.1 / -1.5 | 88.5 / -0.3 | 0.167 | 0.227 |
| | $\omega_{\text{aff}}{=}1.7, \omega_{\text{inv}}{=}1.2$ | 99.2 / +6.8 | 76.9 / +1.3 | 91.5 / +2.7 | 0.174 | 0.233 |
| | $\omega_{\text{aff}}{=}2.0, \omega_{\text{inv}}{=}1.2$ | 99.5 / +7.1 | 75.8 / +0.2 | 90.9 / +2.1 | 0.183 | 0.243 |
| | $\omega_{\text{aff}}{=}2.5, \omega_{\text{inv}}{=}1.2$ | 99.9 / +7.5 | 74.9 / -0.7 | 90.1 / +1.3 | 0.199 | 0.260 |
| | $\omega_{\text{aff}}{=}3.0, \omega_{\text{inv}}{=}1.2$ | 100.0 / +7.6 | 74.5 / -1.1 | 89.4 / +0.6 | 0.216 | 0.276 |
| | $\omega{=}1.0$ | 95.1 / +0.0 | 65.4 / +0.0 | 90.4 / +0.0 | 0.135 | 0.191 |
| | $\omega{=}1.2$ | 97.6 / +2.5 | 69.9 / +4.5 | 93.3 / +2.9 | 0.135 | 0.190 |
| | $\omega{=}1.5$ | 98.9 / +3.8 | 73.9 / +8.5 | 96.2 / +5.8 | 0.155 | 0.209 |
| | $\omega{=}1.7$ | 99.3 / +4.2 | 76.3 / +10.9 | 97.5 / +7.1 | 0.171 | 0.223 |
| | $\omega{=}2.0$ | 99.6 / +4.5 | 80.5 / +15.1 | 98.2 / +7.8 | 0.193 | 0.242 |
| | $\omega{=}2.5$ | 99.7 / +4.6 | 86.1 / +20.7 | 99.2 / +8.8 | 0.229 | 0.271 |
| do(race) | $\omega{=}3.0$ | 99.8 / +4.7 | 90.1 / +24.7 | 99.4 / +9.0 | 0.256 | 0.292 |
| | $\omega_{\text{aff}}{=}1.2, \omega_{\text{inv}}{=}1.0$ | 95.4 / +0.3 | 69.6 / +4.2 | 90.3 / -0.1 | 0.141 | 0.198 |
| | $\omega_{\text{aff}}{=}1.5, \omega_{\text{inv}}{=}1.0$ | 94.8 / -0.3 | 75.5 / +10.1 | 90.0 / -0.4 | 0.147 | 0.203 |
| | $\omega_{\text{aff}}{=}1.7, \omega_{\text{inv}}{=}1.2$ | 97.2 / +2.1 | 78.3 / +12.9 | 92.4 / +2.0 | 0.153 | 0.208 |
| | $\omega_{\text{aff}}{=}2.0, \omega_{\text{inv}}{=}1.2$ | 96.4 / +1.3 | 82.8 / +17.4 | 92.0 / +1.6 | 0.160 | 0.215 |
| | $\omega_{\text{aff}}{=}2.5, \omega_{\text{inv}}{=}1.2$ | 95.6 / +0.5 | 89.0 / +23.6 | 91.7 / +1.3 | 0.178 | 0.231 |
| | $\omega_{\text{aff}}{=}3.0, \omega_{\text{inv}}{=}1.2$ | 94.0 / -1.1 | 92.7 / +27.3 | 91.7 / +1.3 | 0.197 | 0.249 |
| | $\omega{=}1.0$ | 94.6 / +0.0 | 78.3 / +0.0 | 80.8 / +0.0 | 0.134 | 0.193 |
| | $\omega{=}1.2$ | 97.2 / +2.6 | 81.2 / +2.9 | 85.7 / +4.9 | 0.136 | 0.194 |
| | $\omega{=}1.5$ | 98.9 / +4.3 | 83.8 / +5.5 | 90.6 / +9.8 | 0.153 | 0.210 |
| | $\omega{=}1.7$ | 99.5 / +4.9 | 85.7 / +7.4 | 92.9 / +12.1 | 0.165 | 0.222 |
| | $\omega{=}2.0$ | 99.7 / +5.1 | 88.5 / +10.2 | 95.0 / +14.2 | 0.184 | 0.239 |
| | $\omega{=}2.5$ | 99.8 / +5.2 | 91.9 / +13.6 | 97.7 / +16.9 | 0.215 | 0.264 |
| do(finding) | $\omega{=}3.0$ | 99.9 / +5.3 | 93.8 / +15.5 | 98.6 / +17.8 | 0.244 | 0.287 |
| | $\omega_{\text{aff}}{=}1.2, \omega_{\text{inv}}{=}1.0$ | 95.1 / +0.5 | 78.1 / -0.2 | 85.6 / +4.8 | 0.142 | 0.202 |
| | $\omega_{\text{aff}}{=}1.5, \omega_{\text{inv}}{=}1.0$ | 94.9 / +0.3 | 77.8 / -0.5 | 92.0 / +11.2 | 0.141 | 0.201 |
| | $\omega_{\text{aff}}{=}1.7, \omega_{\text{inv}}{=}1.2$ | 97.1 / +2.5 | 80.6 / +2.3 | 93.5 / +12.7 | 0.150 | 0.209 |
| | $\omega_{\text{aff}}{=}2.0, \omega_{\text{inv}}{=}1.2$ | 96.9 / +2.3 | 80.2 / +1.9 | 96.6 / +15.8 | 0.149 | 0.209 |
| | $\omega_{\text{aff}}{=}2.5, \omega_{\text{inv}}{=}1.2$ | 96.4 / +1.8 | 80.1 / +1.8 | 98.8 / +18.0 | 0.151 | 0.212 |
| | $\omega_{\text{aff}}{=}3.0, \omega_{\text{inv}}{=}1.2$ | 95.2 / +0.6 | 79.3 / +1.0 | 99.6 / +18.8 | 0.154 | 0.216 |

Table A.8: MIMIC: Effectiveness (ROC-AUC ↑) and Reversibility (MAE, LPIPS ↓) metrics when changing $\omega_{\text{inv}}$. Increasing $\omega_{\text{inv}}$ consistently increases effectiveness on invariant variables, while degrading intervention effectiveness. When $\omega_{\text{inv}}{=}2.5$, the amplification on invariant attributes becomes comparable to that of the global CFG setting with $\omega{=}2.5$.

| do(key) | Guidance configuration | Sex AUC/Δ | Race AUC/Δ | Finding AUC/Δ | MAE | LPIPS |
|---|---|---|---|---|---|---|
| | $\omega{=}1.0$ | 92.4 / +0.0 | 75.6 / +0.0 | 88.8 / +0.0 | 0.146 | 0.202 |
| | $\omega{=}2.5$ | 99.8 / +7.4 | 90.5 / +14.9 | 99.0 / +10.2 | 0.239 | 0.284 |
| | $\omega_{\text{aff}}{=}2.5, \omega_{\text{inv}}{=}1.0$ | 99.9 / +7.5 | 71.3 / -4.3 | 86.2 / -2.6 | 0.200 | 0.261 |
| do(sex) | $\omega_{\text{aff}}{=}2.5, \omega_{\text{inv}}{=}1.2$ | 99.9 / +7.5 | 74.9 / -0.7 | 90.1 / +1.3 | 0.199 | 0.260 |
| | $\omega_{\text{aff}}{=}2.5, \omega_{\text{inv}}{=}1.5$ | 99.8 / +7.4 | 80.1 / +4.5 | 94.2 / +5.4 | 0.207 | 0.264 |
| | $\omega_{\text{aff}}{=}2.5, \omega_{\text{inv}}{=}1.7$ | 99.7 / +7.3 | 83.2 / +7.6 | 95.9 / +7.1 | 0.214 | 0.269 |
| | $\omega_{\text{aff}}{=}2.5, \omega_{\text{inv}}{=}2.0$ | 99.7 / +7.3 | 86.7 / +11.1 | 97.5 / +8.7 | 0.227 | 0.278 |
| | $\omega_{\text{aff}}{=}2.5, \omega_{\text{inv}}{=}2.5$ | 99.6 / +7.2 | 90.3 / +14.7 | 98.9 / +10.1 | 0.249 | 0.293 |
| | $\omega{=}1.0$ | 95.1 / +0.0 | 65.4 / +0.0 | 90.4 / +0.0 | 0.135 | 0.191 |
| | $\omega{=}2.5$ | 99.7 / +4.6 | 86.1 / +20.7 | 99.2 / +8.8 | 0.229 | 0.271 |
| | $\omega_{\text{aff}}{=}2.5, \omega_{\text{inv}}{=}1.0$ | 91.3 / -3.8 | 89.5 / +24.1 | 88.4 / -2.0 | 0.181 | 0.237 |
| do(race) | $\omega_{\text{aff}}{=}2.5, \omega_{\text{inv}}{=}1.2$ | 95.6 / +0.5 | 89.0 / +23.6 | 91.7 / +1.3 | 0.178 | 0.231 |
| | $\omega_{\text{aff}}{=}2.5, \omega_{\text{inv}}{=}1.5$ | 98.5 / +3.4 | 87.9 / +22.5 | 95.2 / +4.8 | 0.185 | 0.236 |
| | $\omega_{\text{aff}}{=}2.5, \omega_{\text{inv}}{=}1.7$ | 99.2 / +4.1 | 87.4 / +22.0 | 96.8 / +6.4 | 0.191 | 0.242 |
| | $\omega_{\text{aff}}{=}2.5, \omega_{\text{inv}}{=}2.0$ | 99.6 / +4.5 | 86.3 / +20.9 | 98.0 / +7.6 | 0.205 | 0.253 |
| | $\omega_{\text{aff}}{=}2.5, \omega_{\text{inv}}{=}2.5$ | 99.8 / +4.7 | 85.6 / +20.2 | 99.1 / +8.7 | 0.231 | 0.274 |
| | $\omega{=}1.0$ | 94.6 / +0.0 | 78.3 / +0.0 | 80.8 / +0.0 | 0.134 | 0.193 |
| | $\omega{=}2.5$ | 99.8 / +5.2 | 91.9 / +13.6 | 97.7 / +16.9 | 0.215 | 0.264 |
| | $\omega_{\text{aff}}{=}2.5, \omega_{\text{inv}}{=}1.0$ | 93.0 / -1.6 | 77.0 / -1.3 | 99.0 / +18.2 | 0.143 | 0.206 |
| do(finding) | $\omega_{\text{aff}}{=}2.5, \omega_{\text{inv}}{=}1.2$ | 96.4 / +1.8 | 80.1 / +1.8 | 98.8 / +18.0 | 0.151 | 0.212 |
| | $\omega_{\text{aff}}{=}2.5, \omega_{\text{inv}}{=}1.5$ | 98.5 / +3.9 | 84.1 / +5.8 | 98.3 / +17.5 | 0.166 | 0.224 |
| | $\omega_{\text{aff}}{=}2.5, \omega_{\text{inv}}{=}1.7$ | 99.2 / +4.6 | 86.2 / +7.9 | 97.8 / +17.0 | 0.180 | 0.236 |
| | $\omega_{\text{aff}}{=}2.5, \omega_{\text{inv}}{=}2.0$ | 99.6 / +5.0 | 89.4 / +11.1 | 97.1 / +16.3 | 0.202 | 0.254 |
| | $\omega_{\text{aff}}{=}2.5, \omega_{\text{inv}}{=}2.5$ | 99.9 / +5.3 | 92.6 / +14.3 | 95.8 / +15.0 | 0.239 | 0.282 |

## F.3    EXTRA VISUAL RESULTS FOR MIMIC

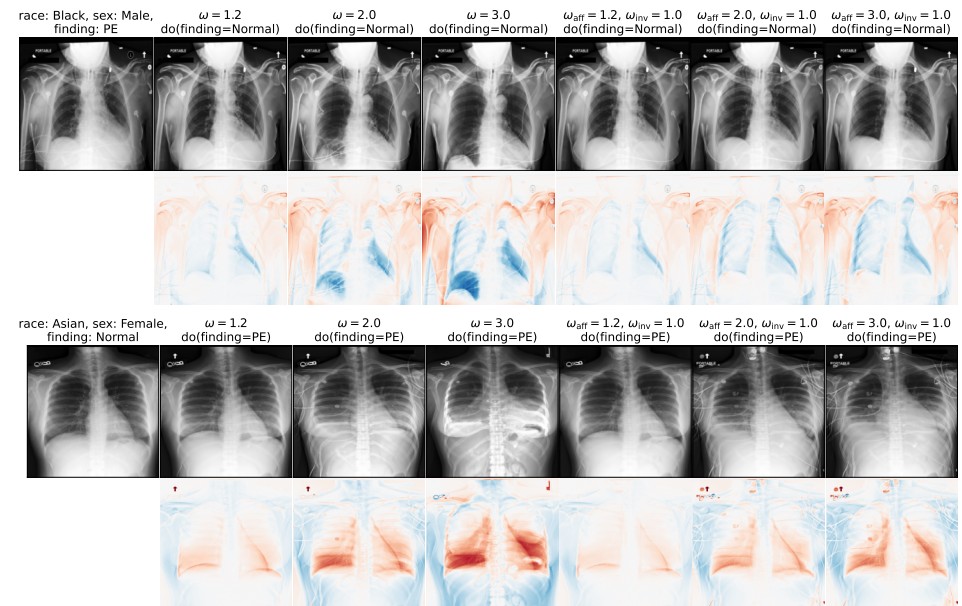

Figure A.15: **Additional qualitative results for `do(finding)` on MIMIC.** Each row shows the original image followed by counterfactuals generated using global CFG ($\omega$) and DCFG ($\omega_{\text{aff}}, \omega_{\text{inv}}$). DCFG better preserves *invariant* attributes and identity while accurately applying the intended intervention. Compared to standard CFG, DCFG produces counterfactuals with more localized changes and stronger identity preservation.

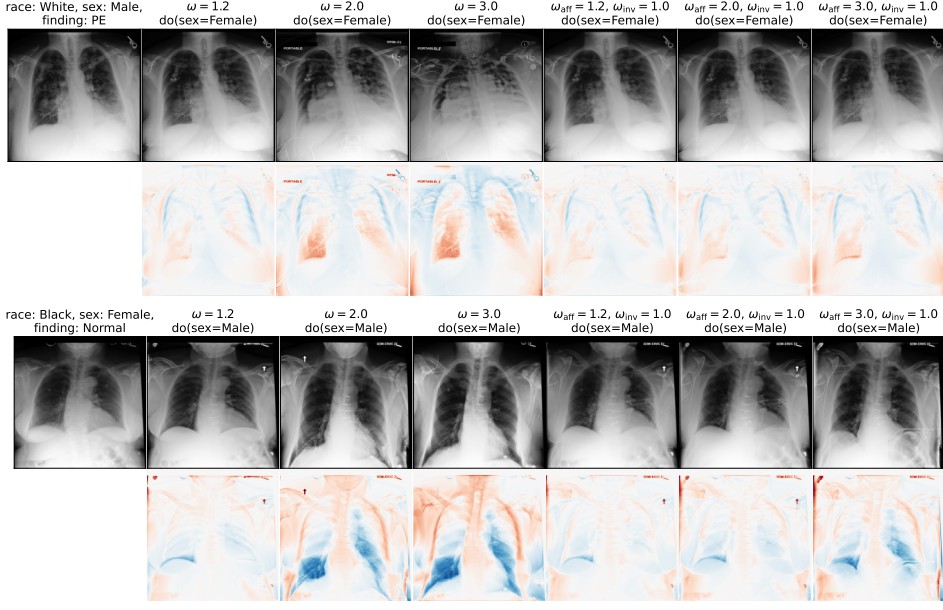

Figure A.16: **Additional qualitative results for `do(sex)` on MIMIC.** Each row shows the original image followed by counterfactuals generated using global CFG ($\omega$) and DCFG ($\omega_{\text{aff}}, \omega_{\text{inv}}$). DCFG better preserves *invariant* attributes and identity while accurately applying the intended intervention on `sex`. Compared to standard CFG, which tends to amplify unrelated features such as disease (i.e. `finding`), DCFG produces counterfactuals with more localized, semantically aligned changes and stronger identity preservation.

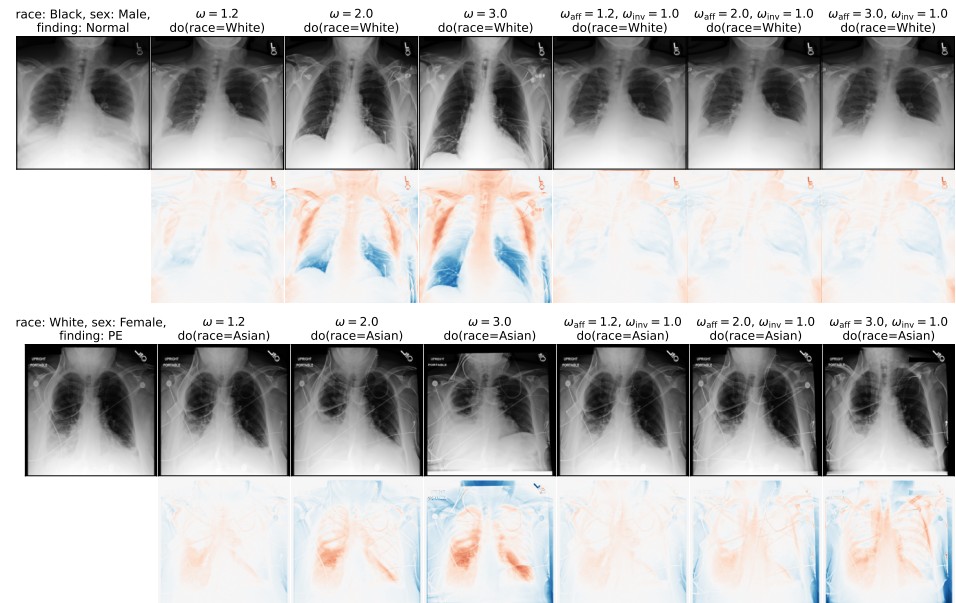

Figure A.17: **Additional qualitative results for do(`race`) on MIMIC.** Each row shows the original image followed by counterfactuals generated using global CFG ($\omega$) and DCFG ($\omega_{\text{aff}}, \omega_{\text{inv}}$). While race interventions correspond to relatively subtle visual changes, standard CFG often amplifies unrelated features such as disease appearance (e.g., `finding`). In contrast, DCFG better preserves *invariant* attributes and identity, producing counterfactuals that are more localized, semantically aligned, and faithful to the intended intervention.

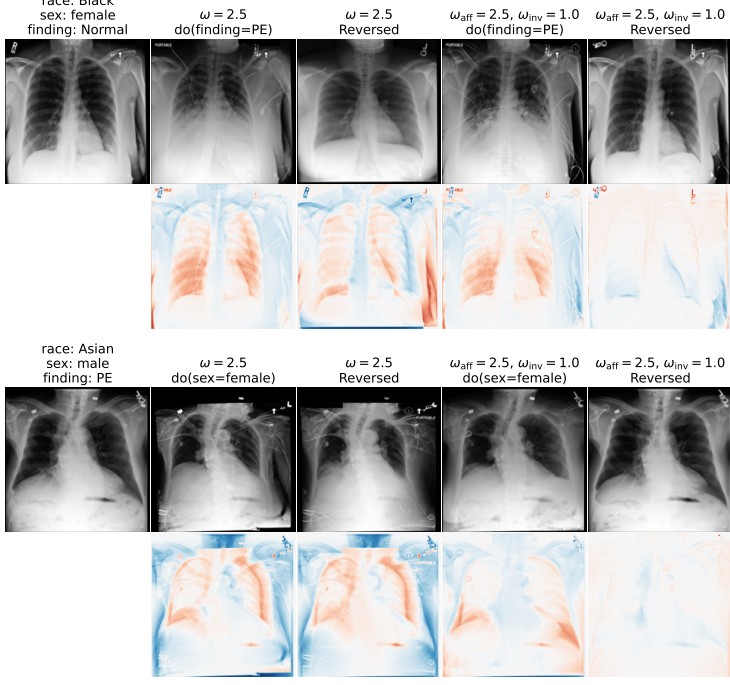

Figure A.18: **Reversibility analysis on MIMIC.** Each row shows the original image, the counterfactual generated using global CFG ($\omega$) or DCFG ($\omega_{\text{int}}, \omega_{\text{inv}}$), their corresponding reversed generations, and the associated difference maps (counterfactual - input, and reversed - input). DCFG more faithfully preserves non-intervened attributes and leads to smaller residuals in the difference maps, indicating better identity preservation.

## G  MULTI-ATTRIBUTE INTERVENTIONS

To demonstrate the generality of the proposed DCFG, we conduct experiments with multi-attribute interventions, i.e., interventions that involve modifying multiple attributes simultaneously. Such interventions can be handled under the two-group partition defined in section 3.3. We also explore an attribute-wise guidance scheme to further highlight the flexibility and generality of DCFG.

Recall that Proposition 1 only requires that different groups are mutually independent given the latent variable. In the case of CelebA, the attributes `Smiling`, `Male`, and `Young` are assumed to be conditionally independent of each other (see Section D.1). This independence allows us to treat each attribute as its own group, thereby extending the two-group partition introduced in Section 3.3 to an attribute-wise setting. In this scheme, each attribute is assigned its own guidance weight (e.g., $\omega_s$ for `Smiling`, $\omega_m$ for `Male`, and $\omega_y$ for `Young`), enabling fine-grained and disentangled control over multi-attribute interventions. However, attribute-wise DCFG is more computationally demanding, as evident from eq. 14, which requires evaluating $\epsilon_\theta$ once for the unconditional case and once per group. This results in $M + 1$ forward passes (where $M$ is the number of groups), compared to 2 for global CFG and 3 for the two-group DCFG. In the following, we present experimental results with two-attribute interventions in Section G.1 and with three-attribute interventions in Section G.2.

### G.1  TWO-ATTRIBUTE INTERVENTIONS

We begin with two-attribute interventions, where two of the variables `Smiling`, `Male`, and `Young` are intervened upon simultaneously. Tables A.9, A.10, and A.11 report the Effectiveness (AUC) and Reversibility (MAE, LPIPS) metrics. Across all pairs, global guidance ($\omega$=2.5) yields high effectiveness for the intervened attributes but also amplifies the non-intervened one. Two-group DCFG ($\omega_{\text{aff}}$=2.5, $\omega_{\text{inv}}$=1.0) consistently suppresses such spurious changes while maintaining high effectiveness on the intervened attributes. DCFG further demonstrates its flexibility and generality through the attribute-wise configuration, where each attribute receives its own guidance weight. This allows selective adjustment of individual attributes, while symmetric settings (e.g., $\omega_s$=$\omega_y$=2.5, $\omega_m$=1.0) recover the group-wise performance. Qualitative examples in Figs. A.19, A.20, and A.21 support these findings, showing that attribute-wise DCFG allows finer control over the intervened attributes and reproduces the outcomes of two-group DCFG under symmetric configurations.

Table A.9: CelebA: Effectiveness (AUC ↑) and Reversibility (MAE, LPIPS ↓) metrics for do(`Smiling`, `Male`). Global CFG ($\omega$=2.5) achieves high effectiveness on both `Smiling` and `Male`, but also amplifies the non-intervened attribute Young. Group-wise DCFG ($\omega_{\text{aff}}$, $\omega_{\text{inv}}$) mitigates this amplification while maintaining high effectiveness on the intervened attributes. Attribute-wise guidance ($\omega_s$ for `Smiling`, $\omega_m$ for `Male`, and $\omega_y$ for `Young`) demonstrates the flexibility and generality of DCFG: changing only one weight selectively affects the corresponding attribute, while setting $\omega_s$=$\omega_m$=2.5 and $\omega_y$=1.0 recovers the group-wise configuration ($\omega_{\text{aff}}$=2.5, $\omega_{\text{inv}}$=1.0).

| Guidance configuration | Smiling AUC/$\Delta$ | Male AUC/$\Delta$ | Young AUC/$\Delta$ | MAE | LPIPS |
|---|---|---|---|---|---|
| $\omega$=1.0 | 83.3 / +0.00 | 90.7 / +0.0 | 79.3 / +0.0 | 0.117 | 0.082 |
| $\omega$=2.5 | 97.7 / +14.4 | 99.5 / +8.8 | 87.7 / +8.4 | 0.227 | 0.155 |
| $\omega_{\text{aff}}$=2.5, $\omega_{\text{inv}}$=1.0 | 98.9 / +15.6 | 99.0 / +8.3 | 72.9 / -6.4 | 0.189 | 0.123 |
| $\omega_s$=1.0, $\omega_m$=1.0, $\omega_y$=1.0 | 82.1 / -1.20 | 85.5 / -5.2 | 81.1 / +1.8 | 0.144 | 0.102 |
| $\omega_s$=1.0, $\omega_m$=2.5, $\omega_y$=1.0 | 79.5 / -3.80 | 99.4 / +8.7 | 76.1 / -3.2 | 0.171 | 0.120 |
| $\omega_s$=2.5, $\omega_m$=1.0, $\omega_y$=1.0 | 99.3 / +16.0 | 82.4 / -8.3 | 77.3 / -2.0 | 0.172 | 0.119 |
| $\omega_s$=2.5, $\omega_m$=2.0, $\omega_y$=1.0 | 98.7 / +15.4 | 96.3 / +5.6 | 74.6 / -4.7 | 0.175 | 0.114 |
| $\omega_s$=2.5, $\omega_m$=2.5, $\omega_y$=1.0 | 98.4 / +15.1 | 98.7 / +8.0 | 72.4 / -6.9 | 0.186 | 0.121 |
| $\omega_s$=2.5, $\omega_m$=3.0, $\omega_y$=1.0 | 97.6 / +14.3 | 99.5 / +8.8 | 71.0 / -8.3 | 0.198 | 0.128 |

Table A.10: CelebA: Effectiveness (AUC ↑) and Reversibility (MAE, LPIPS ↓) metrics for do(Smiling, Young). Global CFG ($\omega$=2.5) achieves high effectiveness on both Smiling and Young but also amplifies the non-intervened attribute Male. Group-wise DCFG ($\omega_{\text{aff}}, \omega_{\text{inv}}$) mitigates this amplification while maintaining high effectiveness on the intervened attributes. Attribute-wise guidance ($\omega_{\text{s}}$ for Smiling, $\omega_{\text{m}}$ for Male, and $\omega_{\text{y}}$ for Young) demonstrates the flexibility and generality of DCFG: changing only one weight selectively affects the corresponding attribute, while setting $\omega_s$=$\omega_y$=2.5 and $\omega_m$=1.0 recovers the group-wise configuration ($\omega_{\text{aff}}$=2.5, $\omega_{\text{inv}}$=1.0).

| Guidance configuration | Smiling AUC/$\Delta$ | Male AUC/$\Delta$ | Young AUC/$\Delta$ | MAE | LPIPS |
|---|---|---|---|---|---|
| $\omega$=1.0 | 83.6 / +0.0 | 94.6 / +0.0 | 60.1 / +0.0 | 0.123 | 0.094 |
| $\omega$=2.5 | 96.8 / +13.2 | 99.8 / +5.2 | 75.7 / +15.6 | 0.221 | 0.138 |
| $\omega_{\text{aff}}$=2.5, $\omega_{\text{inv}}$=1.0 | 97.5 / +13.9 | 85.5 / -9.1 | 79.0 / +18.9 | 0.204 | 0.148 |
| $\omega_{\text{s}}$=1.0, $\omega_{\text{m}}$=1.0, $\omega_{\text{y}}$=1.0 | 82.1 / -1.5 | 93.8 / -0.8 | 58.1 / -2.0 | 0.139 | 0.107 |
| $\omega_{\text{s}}$=1.0, $\omega_{\text{m}}$=1.0, $\omega_{\text{y}}$=2.5 | 77.7 / -5.9 | 85.6 / -9.0 | 84.2 / +24.1 | 0.176 | 0.137 |
| $\omega_{\text{s}}$=2.5, $\omega_{\text{m}}$=1.0, $\omega_{\text{y}}$=1.0 | 98.9 / +15.3 | 91.5 / -3.1 | 54.9 / -5.2 | 0.173 | 0.125 |
| $\omega_{\text{s}}$=2.5, $\omega_{\text{m}}$=1.0, $\omega_{\text{y}}$=2.0 | 97.9 / +14.3 | 87.2 / -7.4 | 71.0 / +10.9 | 0.189 | 0.138 |
| $\omega_{\text{s}}$=2.5, $\omega_{\text{m}}$=1.0, $\omega_{\text{y}}$=2.5 | 97.0 / +13.4 | 86.1 / -8.5 | 77.9 / +17.8 | 0.201 | 0.147 |
| $\omega_{\text{s}}$=2.5, $\omega_{\text{m}}$=1.0, $\omega_{\text{y}}$=3.0 | 96.1 / +12.5 | 83.7 / -10.9 | 84.4 / +24.3 | 0.212 | 0.154 |

Table A.11: CelebA: Effectiveness (AUC ↑) and Reversibility (MAE, LPIPS ↓) metrics for do(Male, Young). Global CFG ($\omega$=2.5) achieves high effectiveness on both Male and Young but also amplifies the non-intervened attribute Smiling. Group-wise DCFG ($\omega_{\text{aff}}, \omega_{\text{inv}}$) mitigates this amplification while maintaining high effectiveness on the intervened attributes. Attribute-wise guidance ($\omega_{\text{s}}$ for Smiling, $\omega_{\text{m}}$ for Male, and $\omega_{\text{y}}$ for Young) demonstrates the flexibility and generality of DCFG: changing only one weight selectively affects the corresponding attribute, while setting $\omega_m$=$\omega_y$=2.5 and $\omega_s$=1.0 recovers the group-wise configuration ($\omega_{\text{aff}}$=2.5, $\omega_{\text{inv}}$=1.0).

| Guidance configuration | Smiling AUC/$\Delta$ | Male AUC/$\Delta$ | Young AUC/$\Delta$ | MAE | LPIPS |
|---|---|---|---|---|---|
| $\omega$=1.0 | 82.5 / +0.0 | 89.1 / +0.0 | 63.1 / +0.0 | 0.122 | 0.088 |
| $\omega$=2.5 | 98.5 / +16.0 | 99.6 / +10.5 | 79.8 / +16.7 | 0.216 | 0.143 |
| $\omega_{\text{aff}}$=2.5, $\omega_{\text{inv}}$=1.0 | 80.0 / -2.5 | 99.2 / +10.1 | 83.9 / +20.8 | 0.198 | 0.144 |
| $\omega_{\text{s}}$=1.0, $\omega_{\text{m}}$=1.0, $\omega_{\text{y}}$=1.0 | 85.2 / +2.7 | 88.1 / -1.0 | 63.4 / +0.3 | 0.154 | 0.114 |
| $\omega_{\text{s}}$=1.0, $\omega_{\text{m}}$=1.0, $\omega_{\text{y}}$=2.5 | 82.8 / +0.3 | 86.5 / -2.6 | 86.1 / +23.0 | 0.183 | 0.137 |
| $\omega_{\text{s}}$=1.0, $\omega_{\text{m}}$=2.5, $\omega_{\text{y}}$=1.0 | 83.1 / +0.6 | 99.4 / +10.3 | 65.6 / +2.5 | 0.177 | 0.126 |
| $\omega_{\text{s}}$=1.0, $\omega_{\text{m}}$=2.5, $\omega_{\text{y}}$=2.0 | 80.5 / -2.0 | 99.3 / +10.2 | 76.5 / +13.4 | 0.183 | 0.128 |
| $\omega_{\text{s}}$=1.0, $\omega_{\text{m}}$=2.5, $\omega_{\text{y}}$=2.5 | 80.3 / -2.2 | 98.8 / +9.7 | 81.9 / +18.8 | 0.199 | 0.141 |
| $\omega_{\text{s}}$=1.0, $\omega_{\text{m}}$=2.5, $\omega_{\text{y}}$=3.0 | 82.0 / -0.5 | 98.4 / +9.3 | 85.0 / +21.9 | 0.208 | 0.147 |

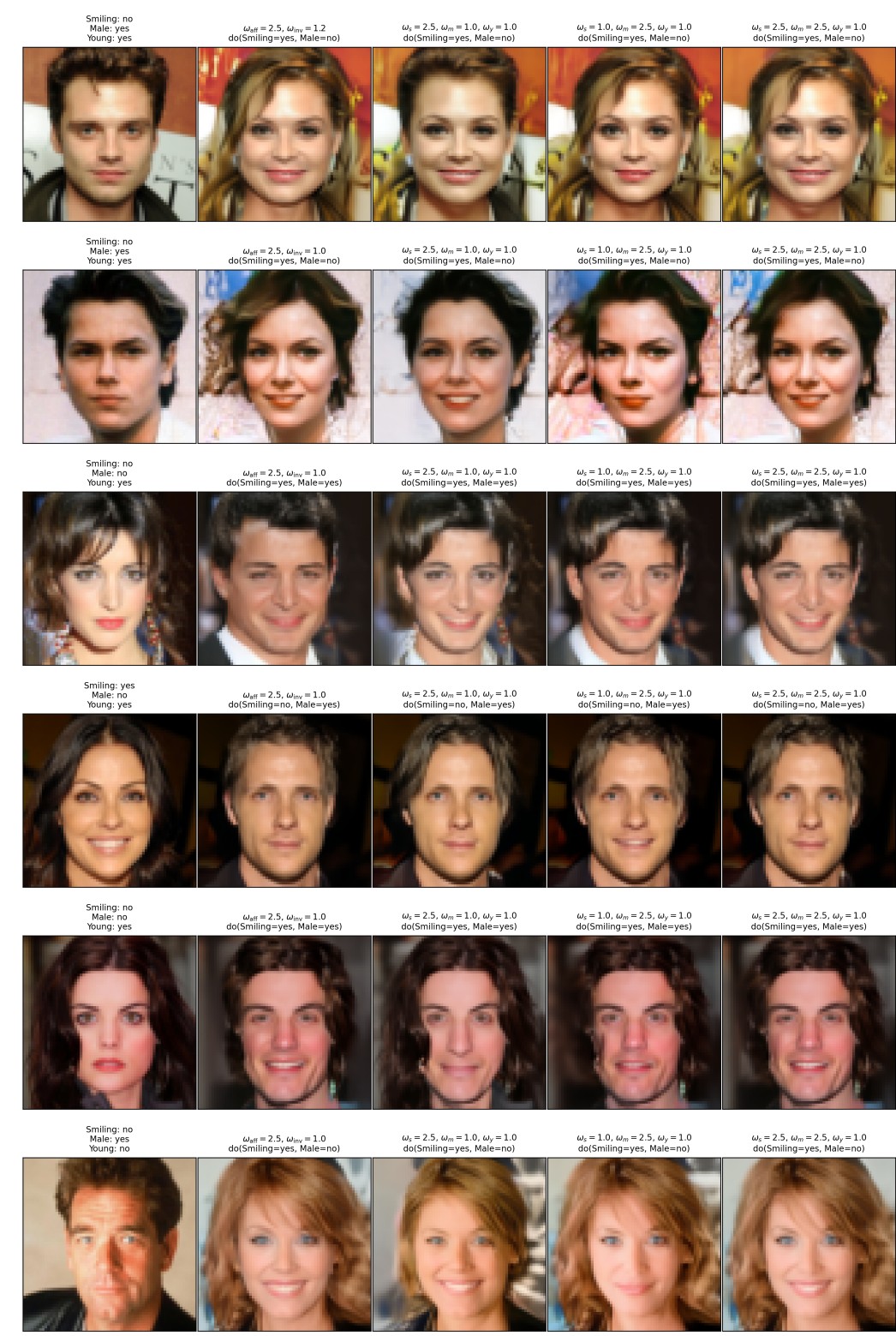

Figure A.19: **Qualitative results for do(Smiling, Male) on CelebA-HQ.** Each row shows the original image followed by counterfactuals generated with two-group DCFG ($\omega_{\mathrm{aff}}, \omega_{\mathrm{inv}}$) and with attribute-wise DCFG ($\omega_{\mathrm{s}}$ for Smiling, $\omega_{\mathrm{m}}$ for Male, and $\omega_{\mathrm{y}}$ for Young). Attribute-wise DCFG provides more flexible configurations, allowing selective control of individual attributes while recovering the two-group DCFG results under symmetric settings.

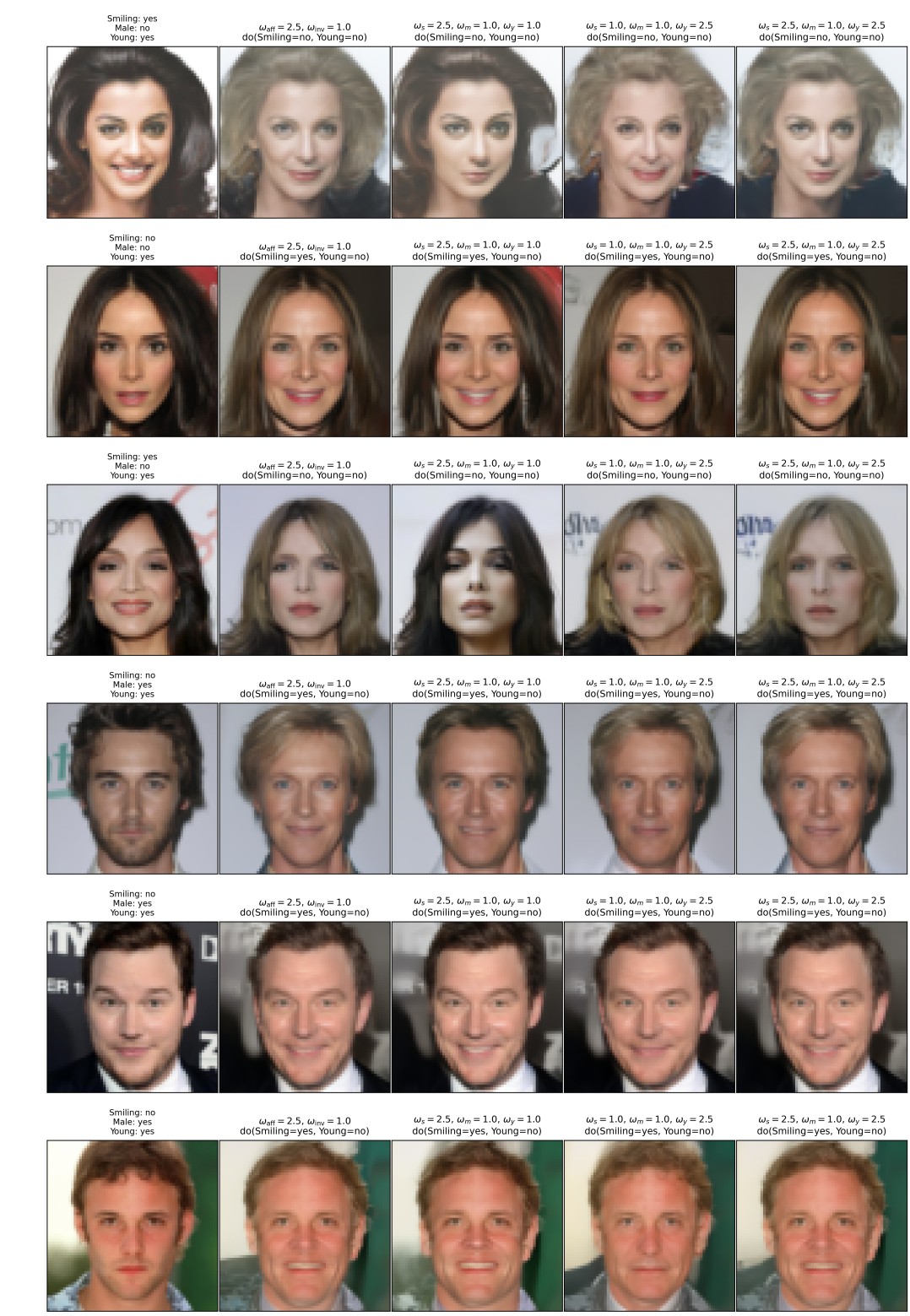

Figure A.20: **Qualitative results for `do(Smiling, Young)` on CelebA-HQ.** Each row shows the original image followed by counterfactuals generated with two-group DCFG ($\omega_{\mathrm{aff}}, \omega_{\mathrm{inv}}$) and with attribute-wise DCFG ($\omega_{\mathrm{s}}$ for Smiling, $\omega_{\mathrm{m}}$ for Male, and $\omega_{\mathrm{y}}$ for Young). Attribute-wise DCFG provides more flexible configurations, allowing selective control of individual attributes while recovering the two-group DCFG results under symmetric settings.

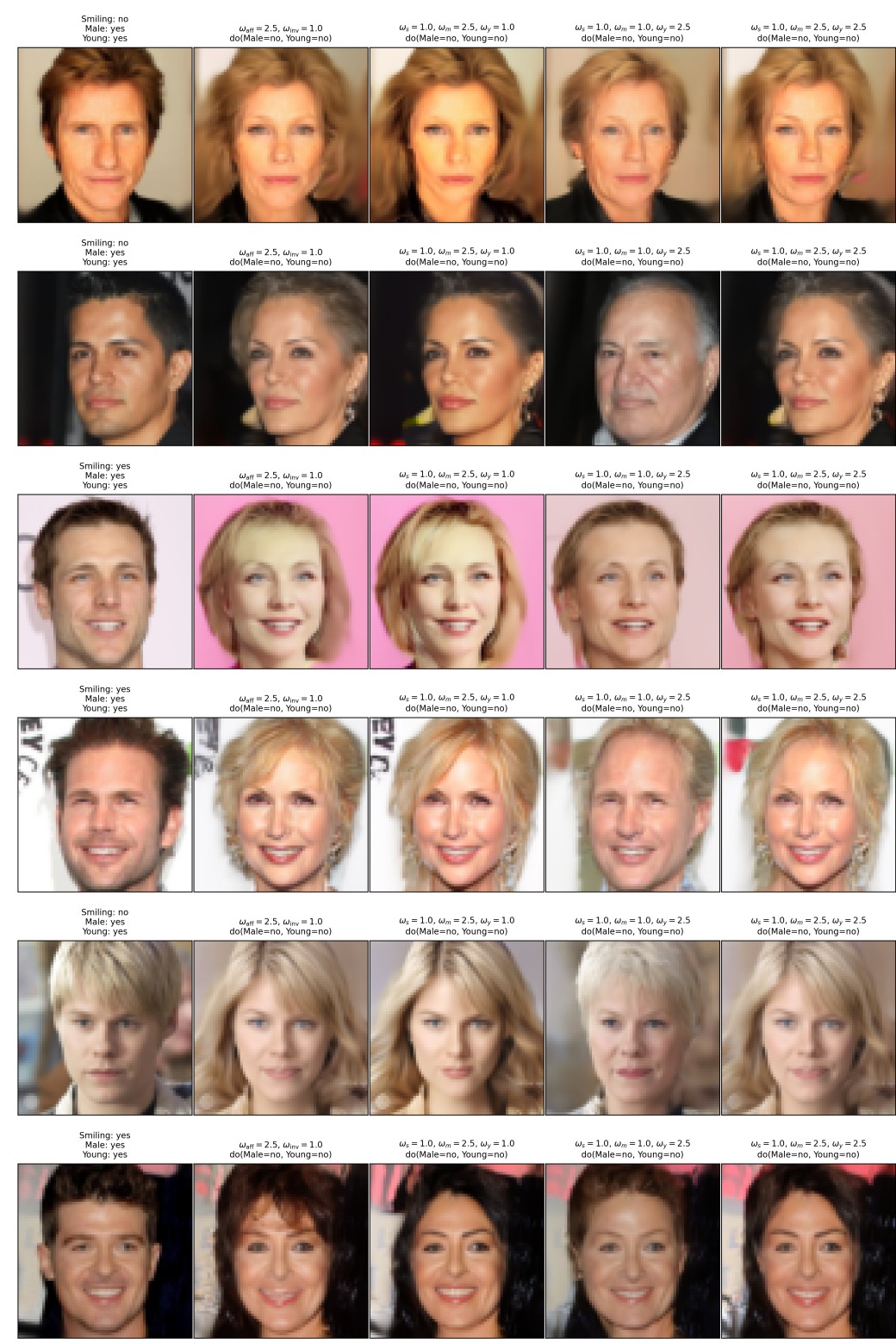

Figure A.21: **Qualitative results for `do(Male, Young)` on CelebA-HQ.** Each row shows the original image followed by counterfactuals generated with two-group DCFG ($\omega_{\text{aff}}, \omega_{\text{inv}}$) and with attribute-wise DCFG ($\omega_{\text{s}}$ for `Smiling`, $\omega_{\text{m}}$ for `Male`, and $\omega_{\text{y}}$ for `Young`). Attribute-wise DCFG provides more flexible configurations, allowing selective control of individual attributes while recovering the two-group DCFG results under symmetric settings.

## G.2 THREE-ATTRIBUTE INTERVENTIONS

We then move to three-attribute interventions, where all of Smiling, Male, and Young are intervened simultaneously. Table A.12 reports the corresponding Effectiveness (AUC) and Reversibility (MAE, LPIPS) metrics. Notably, in this setting all attributes are intervened, which makes global CFG and two-group DCFG identical, as no invariant attributes remain. In this setting, attribute-wise DCFG provides additional flexibility: it enables selective control of the guidance strength across attributes, while symmetric settings (e.g., $\omega_s=\omega_m=\omega_y=2.5$) recover the outcomes of the global/two-group configuration ($\omega=2.5^*$). Qualitative examples in Fig. A.22 further illustrate this flexibility, showing that attribute-wise DCFG can selectively control each attribute under the all-attribute intervention.

Table A.12: **CelebA: Effectiveness (AUC ↑) and Reversibility (MAE, LPIPS ↓) metrics for do(Smiling, Male, Young).** Since all three attributes are intervened, there are no invariant attributes remaining; hence, global CFG ($\omega$) and two-group DCFG ($\omega_{\mathrm{aff}}, \omega_{\mathrm{inv}}$) become identical, as the same guidance weight is applied to every attribute. Attribute-wise DCFG ($\omega_{\mathrm{s}}$ for Smiling, $\omega_{\mathrm{m}}$ for Male, and $\omega_{\mathrm{y}}$ for Young) demonstrates the flexibility and generality of DCFG by enabling selective adjustment of each attribute. In particular, setting $\omega_s=\omega_m=\omega_y=2.5$ recovers the global/two-group configuration ($\omega=2.5^*$), where $^*$ denotes the equivalence between global CFG and two-group DCFG in this all-attribute intervention setting.

| Guidance configuration | Smiling AUC/$\Delta$ | Male AUC/$\Delta$ | Young AUC/$\Delta$ | MAE | LPIPS |
|---|---|---|---|---|---|
| $\omega=1.0$ | 86.1 / +0.0 | 88.4 / +0.0 | 64.0 / +0.0 | 0.124 | 0.093 |
| $\omega=2.5^*$ | 98.1 / +12.0 | 99.0 / +10.6 | 84.7 / +20.7 | 0.207 | 0.138 |
| $\omega_{\mathrm{s}}=1.0, \omega_{\mathrm{m}}=1.0, \omega_{\mathrm{y}}=1.0$ | 84.1 / -2.0 | 88.6 / +0.2 | 66.4 / +2.4 | 0.151 | 0.114 |
| $\omega_{\mathrm{s}}=2.5, \omega_{\mathrm{m}}=1.0, \omega_{\mathrm{y}}=1.0$ | 99.3 / +13.2 | 86.8 / -1.6 | 66.3 / +2.3 | 0.176 | 0.123 |
| $\omega_{\mathrm{s}}=1.0, \omega_{\mathrm{m}}=2.5, \omega_{\mathrm{y}}=1.0$ | 80.9 / -5.2 | 99.3 / +10.9 | 64.8 / +0.8 | 0.183 | 0.130 |
| $\omega_{\mathrm{s}}=1.0, \omega_{\mathrm{m}}=1.0, \omega_{\mathrm{y}}=2.5$ | 79.1 / -7.0 | 86.6 / -1.8 | 88.5 / +24.5 | 0.181 | 0.141 |
| $\omega_{\mathrm{s}}=1.0, \omega_{\mathrm{m}}=2.5, \omega_{\mathrm{y}}=2.5$ | 79.0 / -7.1 | 99.2 / +10.8 | 83.8 / +19.8 | 0.189 | 0.142 |
| $\omega_{\mathrm{s}}=2.5, \omega_{\mathrm{m}}=1.0, \omega_{\mathrm{y}}=2.5$ | 97.8 / +11.7 | 86.1 / -2.3 | 85.9 / +21.9 | 0.188 | 0.147 |
| $\omega_{\mathrm{s}}=2.5, \omega_{\mathrm{m}}=2.5, \omega_{\mathrm{y}}=1.0$ | 98.4 / +12.3 | 98.4 / +10.0 | 67.7 / +3.7 | 0.191 | 0.126 |
| $\omega_{\mathrm{s}}=2.5, \omega_{\mathrm{m}}=2.5, \omega_{\mathrm{y}}=2.0$ | 97.9 / +11.8 | 98.5 / +10.1 | 77.6 / +13.6 | 0.198 | 0.133 |
| $\omega_{\mathrm{s}}=2.5, \omega_{\mathrm{m}}=2.5, \omega_{\mathrm{y}}=2.5$ | 97.7 / +11.6 | 98.3 / +9.9 | 81.6 / +17.6 | 0.210 | 0.139 |
| $\omega_{\mathrm{s}}=2.5, \omega_{\mathrm{m}}=2.5, \omega_{\mathrm{y}}=3.0$ | 97.6 / +11.5 | 98.6 / +10.2 | 84.8 / +20.8 | 0.220 | 0.151 |

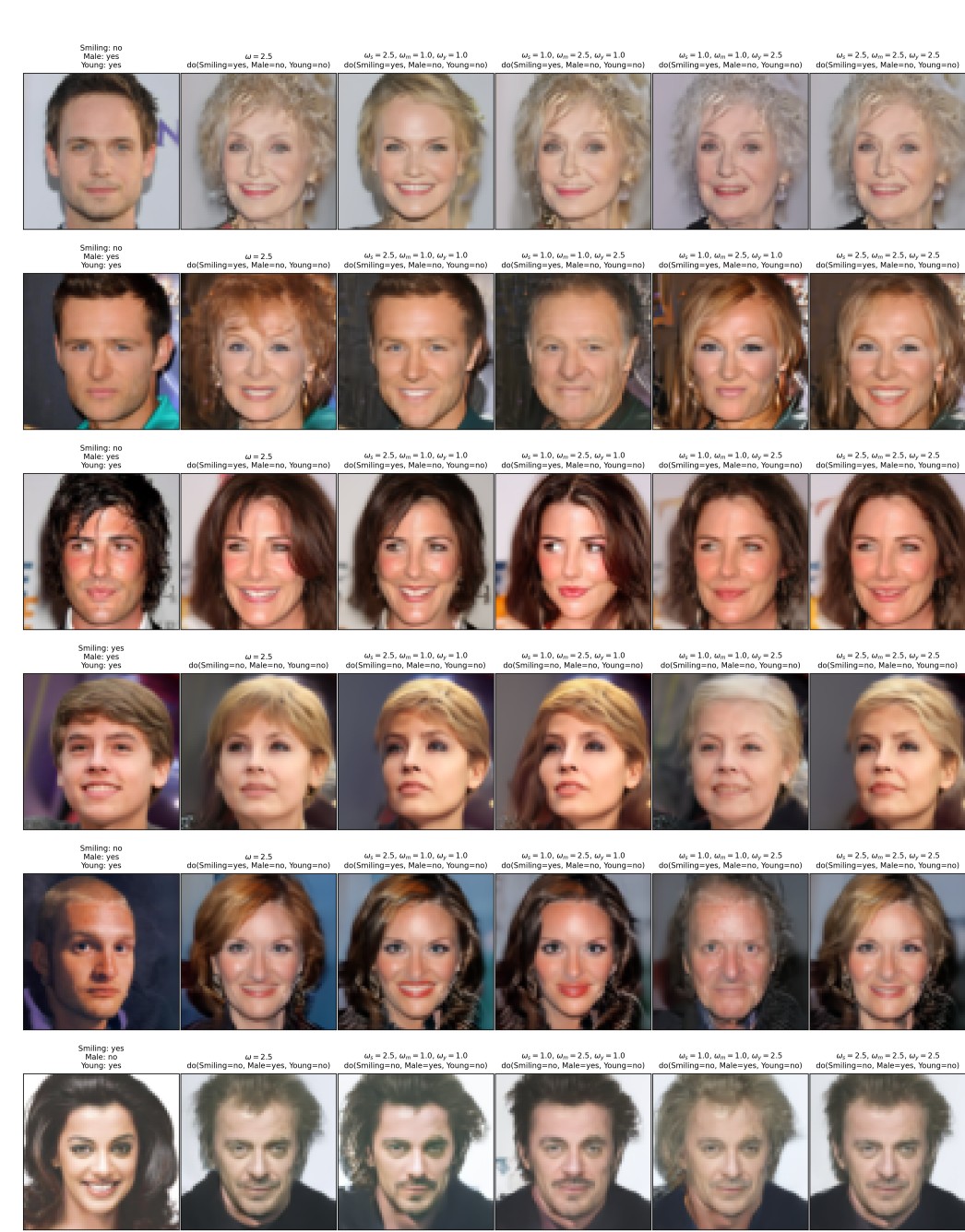

Figure A.22: **Qualitative results for do(Smiling, Male, Young) on CelebA-HQ.** The first column shows the original image, followed by counterfactuals generated with global/two-group DCFG ($\omega$=2.5) and with attribute-wise DCFG ($\omega_{\mathrm{s}}$, $\omega_{\mathrm{m}}$, $\omega_{\mathrm{y}}$). Attribute-wise DCFG enables selective control of the three attributes (e.g., raising only $\omega_{\mathrm{s}}$ or $\omega_{\mathrm{y}}$) while symmetric settings (e.g., $\omega_{\mathrm{s}}$=$\omega_{\mathrm{m}}$=$\omega_{\mathrm{y}}$=2.5) reproduce the outcomes of the global/two-group configuration.

# H    USAGE OF LLM

Portions of the writing in this paper were assisted by a large language model (ChatGPT), specifically for phrasing, grammar improvements, and polishing of text. The research ideas, methods, experiments, and analyses were conceived, implemented, and verified entirely by the authors. All content has been reviewed and verified by the authors, who take full responsibility for the final manuscript.

