# OpenReview forum: "Decoupled Classifier-Free Guidance for Counterfactual Diffusion Models"
_ICLR.cc/2026/Conference — ICLR 2026 Conference Withdrawn Submission_

### Official Review · Reviewer_6N8V · 2025-10-16

**Soundness:** 2
**Presentation:** 3
**Contribution:** 1
**Rating:** 2
**Confidence:** 4

**Summary:**

The authors propose Decoupled Classifier-Free Guidance, a counterfactual image generation method based on classifier-free guidance. The difference to prior work is that the authors cluster attributes into different groups and apply different guidance weights to them. THe method is empirically evaluated on CelebA-HQ, MIMIC-CXR and EMBED.

**Strengths:**

1. The paper is well-written. I appreciate how the approach is contrasted with the existing method by [1]. It becomes immediately clear what is done differently in this work.

2. The experimental evaluation is quite strong. The method is demonstrated on many different data sets with many examples. I am not a radiologist, so I cannot judge the mammograms and chest x-ray images. However, it is clear that this is an important application domain.

3. The proposed method is simple.

[1] Sanchez, Pedro, and Sotirios A. Tsaftaris. "Diffusion causal models for counterfactual estimation." Conference on Causal Learning and Reasoning (2022).

**Weaknesses:**

1. My main concern with this method is that it is somewhat trivial. The only novelty seems to lie in a grouping of attribute variables and then applying different guidance strengths to each group. Given the lack of contribution, I am afraid that this work is far from being publishable.

2. The soundness of this method is also not so clear to me. From a mathematical perspective, the weights $w_i$ should actually all be $1$. Also, the authors propose to have one weight for the "affected" variables and an additional weight for "unaffected" variables and it is not clear why this is a reasonable grouping. Generally, I do not see any deeper grounding to the argument of why we should group attributes and apply different guidance weights to them and why the groups are chosen the way that they are.

3. Some notation could be improved. For instance the $\mathrm{pa}$ notation: In case the authors do not know it, "$\mathrm{pa}$" means "parents", in the sense of "causal parents". I would therefore not use $\mathrm{pa}^{(m)}$ to denote attribute groupings, because it sounds as if we are grouping variables by their causal mechanisms (which is apparently not the case).

**Questions:**

* I highly suggest to remove "proposition 1". This factorization is trivial and there is nothing to be proved, as far as I can see. Maybe I am missing something?

* I do not understand why the method is based on classifier-free guidance. It seems to me that the method could also be implemented with classifier guidance.

* Why is the method called "Decoupled Classifier-Free Guidance"? What is decoupled? It seems more that it is grouped, so should be called something like "Grouped Classifier-Free Guidance".

* In appendix D, it says that the anti-correlation between "young" and "male" stems from data set bias. What is "data set bias"? To me it seems more like selection bias: Given that someone is a celebrity, being young and female are dependent.

---

> ### Author Response · Authors · 2025-11-24
>
> We thank the reviewer for the detailed feedback and for raising several important points regarding the contribution, soundness, and clarity of the method. We appreciate the reviewer’s positive remarks about the quality of writing, the clarity of the comparison to prior work, and the strength of the experimental evaluation across multiple datasets. Below we address each concern in turn.
>
> ***Novelty and simplicity of the method.***
>
> Our main contribution is not only the grouping mechanism itself but the identification and systematic characterization of a persistent amplification failure mode in classifier-free guidance, which has not been formally analyzed or addressed in prior work, despite the widespread use of CFG in counterfactual diffusion. Building on this observation, we aimed to provide a minimal, practical solution that is extremely easy to adopt and requires no retraining or architectural modifications. While simple by design, the method consistently improves counterfactual consistency across datasets and imaging domains. We will clarify this motivation and articulate the contribution more explicitly in a future revision. We also note that the same reviewer listed “The proposed method is simple” as a strength, which aligns with our intention to introduce a lightweight and easily deployable improvement rather than a complex architectural change.
>
> ***Clarification on whether guidance weights “should be 1”.***
>
> We believe the statement that “from a mathematical perspective, the weights should all be 1” reflects a misunderstanding of how classifier-free guidance (CFG) operates.
>
> The original CFG paper (Ho & Salimans, 2022) explicitly states that the guidance weight is a free inference-time hyperparameter, not a quantity determined by the probabilistic model or the diffusion training objective. In their formulation, the model is trained only for w = 1, but at sampling time, the guidance weight is intentionally varied (typically w > 1) to amplify conditional information. The paper describes this as a controllable trade-off between sample fidelity and conditional strength. In other words, the use of w > 1 is not only permitted,  but the intended use of CFG.
>
> If all weights were fixed to 1 in our setting, we would simply recover the conditional model without any guidance, which may provide insufficient control for counterfactual changes, as reflected in our results. Adjusting the weights (including allowing different weights for different semantic groups) is fully consistent with the purpose of classifier-free guidance as described in the original paper.
>
> ***Rationale Behind the Intervened–Invariant Attribute Partition.***
>
> The grouping is grounded both in counterfactual semantics and in the behavior of classifier-free guidance (CFG). A counterfactual query specifies (i) which attributes are intervened on (and their descendants), which are expected to change, and (ii) which attributes define the context and are intended to remain fixed. Using different guidance strengths for these two sets directly reflects this semantics: we want to push the model along the dimensions that should change while stabilizing those that should not. Empirically, our experiments show that if all attributes share a single global guidance weight, standard CFG produces systematic attribute amplification—non-target attributes drift even when they are independent. This failure mode is precisely why differential weighting is necessary. The intervened–invariant partition is therefore not arbitrary; it arises naturally from the structure of counterfactual queries and directly addresses an observed limitation of standard CFG.
>
> ***Notation issues.***
>
> We use “pa” in the standard sense of causal parents and explicitly define it on page 2 (“given its causal parents pa”). In the equations, our intention was only to maintain consistent notation for conditioning variables, not to suggest that the attribute groups correspond to causal parents. We will clarify this distinction in a future version.

---

> ### Author Response · Authors · 2025-11-24
>
> (Continued)
>
> ***Regarding Proposition 1.***
>
> Proposition 1 is intentionally simple; our aim was not to introduce a deep theoretical result but to state the factorization explicitly for completeness and clarity, as it formalizes how the guided score decomposes when conditioning is split across attribute groups. That said, we agree that the result is straightforward, and presenting it in the main text may not be necessary. In a future revision, we will consider moving it to the appendix.
>
> ***Why classifier-free guidance instead of classifier guidance?***
>
> In principle, the idea of decoupling guidance strengths could also be applied to classifier guidance. Our choice of classifier-free guidance (CFG) is primarily practical: CFG does not require training an external classifier on both clean and noisy samples, which can be expensive, unstable, and dataset-dependent, especially in medical domains where labels may be sparse or noisy. CFG allows us to generate counterfactuals directly through the model’s conditioning interface without introducing additional networks or supervision.
>
> More importantly, the amplification issue we address arises specifically from the CFG extrapolation mechanism, where a single global guidance weight pushes all conditioning dimensions simultaneously. This is the mechanism used in most conditional diffusion models today, and it is the setting in which we observed consistent cross-attribute amplification. Our method, therefore, targets the regime in which the problem actually occurs in practice.
>
> That said, the general idea of attributing different guidance strengths to different semantic components could indeed be adapted to a classifier-guided setup. We will clarify this point and discuss the connection to classifier guidance in a future revision.
>
> ***On the name “Decoupled Classifier-Free Guidance.”***
>
> We use the term decoupled because the method explicitly separates the contribution of different parts of the conditioning signal in the CFG update. Standard classifier-free guidance applies a single global weight to the entire conditional embedding, which couples all attribute dimensions together during the extrapolation. Our method breaks this coupling by allowing different guidance weights for different semantic components of the conditioning input (intervened vs. invariant attributes), thereby decoupling their influence in the guidance step.
>
> While this results in grouping attributes for implementation, the key conceptual change is that their effects on the guided score are no longer tied to a single shared scale, but are controlled independently. For this reason, “decoupled” reflects the underlying mechanism more directly than “grouped,” which describes the implementation but not the guidance behavior being modified.
>
> ***On the term “dataset bias.”***
>
> It is sensible that in CelebA, the dependence between “young’’ and “female’’ can be explained by selection effects in how celebrity images are collected, and this can indeed manifest as an apparent causal or demographic association. In our appendix, we used “dataset bias’’ in a broad, informal sense to refer to empirical correlations present in the training data, but we agree that the term “selection bias’’ is more precise in this context. We will revise the wording in a future version to avoid ambiguity.

---

### Official Review · Reviewer_4vnV · 2025-10-30

**Soundness:** 2
**Presentation:** 3
**Contribution:** 2
**Rating:** 2
**Confidence:** 3

**Summary:**

This paper identifies a key limitation in the conventional Classifier-Free Guidance (CFG) approach for counterfactual generation: applying a global guidance weight $\omega$ to the entire counterfactual embedding can violate causal relations and unintentionally alter attributes that should remain unchanged. To address this issue, the author proposes Decoupled Classifier-Free Guidance (DCFG), which employs multiple MLPs, each corresponding to a specific attribute, to generate independent semantic attribute vectors and provide more targeted counterfactual generation guidance. The DCFG framework is evaluated on three datasets, demonstrating its effectiveness over conventional CFG.

**Strengths:**

1. The discussion on how traditional CFG can violate causal relations is interesting and can be inspirational for future research in counterfactual generation.

2. DCFG shows promising performance across all case studies.

3. The paper presents a clear and thorough explanation of the technical background and motivation.

**Weaknesses:**

1. The proposed solution, which uses a separate MLP for each attribute, is not scalable. For instance, while the CelebA-HQ experiments involve only three attributes, the CelebA dataset includes 40 attributes. Applying DCFG to all attributes would require 40 MLPs, which raises a serious concern on the scalability.

2. Related areas such as disentangled representation learning and debiasing for protected attributes have extensively explored methods for isolating and manipulating specific attribute features without affecting others. Although the paper claims to focus on counterfactual inference and structural causal models, its discussion of causality remains limited. Beyond the abduction–action–prediction procedure, there is little discussion to causal graphs or explicit modeling of causal relations.

3. The experiment setup appears closer to studying disentangled representation learning rather than causality, as the attributes used in the case studies (e.g., Figure A.1) are independent with each other rather than causally related. Consequently, the experiments may not sufficiently demonstrate DCFG’s ability to leverage causal relations for counterfactual generation.

**Questions:**

Please refer to the Weaknesses.

---

> ### Author Response · Authors · 2025-11-24
>
> We thank the reviewer for the detailed and thoughtful comments. We appreciate the positive remarks on the technical clarity and motivation, and the recognition that DCFG performs well across all case studies. Below, we address the reviewer’s concerns regarding scalability, relation to disentanglement and causality, and the experimental setup.
>
> ***Scalability of using separate MLPs for each attribute.***
>
> Although using a per-attribute MLP may appear unscalable, each MLP (per attribute) in our implementation is intentionally small. With the current architecture (1→32→32), each sub-MLP contains 1,120 parameters; even with 40 attributes, the total remains only 44,800 parameters, which is still very lightweight. The resulting 32-dimensional embeddings for each attribute are concatenated into a 32×40 = 1280-dimensional conditioning vector, and the embedding size per attribute (i.e., 32) can be adjusted if needed.
>
> By contrast, using a single joint MLP to encode the same 40 attributes into a 1280-dimensional embedding would require a much larger network: a 40→1280→1280 MLP contains about 1.69 million parameters. Thus, the per-attribute design is substantially more parameter-efficient while preserving the desired attribute structure.
>
> We also note that the “3×96” entry in Table A.2 was a typo; the correct conditioning-embedding dimension for CelebA is 32×3=96, and the total number of parameters in our attribute-encoding module for the CelebA experiments is 3,360. We will correct this in a future revision.
>
> ***Relation to disentanglement, debiasing, and causality.***
>
> We agree that related areas, such as disentangled representation learning and debiasing for protected attributes, offer complementary perspectives on isolating and manipulating specific factors. Our work already introduces the intervened–invariant attribute grouping motivated by a causal perspective, but we agree that this connection could be made more explicit. In a future revision, we will clarify how this grouping relates to causal assumptions and expand the discussion on causal graphs and structural causal models to better situate our method within the broader causal generative modeling literature.
>
> ***Experimental setup and causal relevance.***
>
> In our current experiments, we intentionally focused on attribute sets that are independent or only weakly related, as this setting makes unintended cross-attribute amplification easier to observe and evaluate. Our goal was to highlight that standard CFG can induce undesirable changes even for attributes that should be unrelated, and to show that DCFG mitigates this effect. We agree that applying DCFG to settings with richer and explicitly defined causal graphs would further strengthen the causal perspective. We will explore this direction and include experiments with more complex causal structures in a future revision.

---

### Official Review · Reviewer_MDyx · 2025-10-31

**Soundness:** 3
**Presentation:** 3
**Contribution:** 2
**Rating:** 4
**Confidence:** 3

**Summary:**

In this work, the authors address the issue that classifier-free guidance for counterfactual generation can lead to attribution amplification, referring to unwanted correlations between attributes. To mitigate this problem, they propose a new method, decoupled classifier free guidance, which leverages a causal graph. The proposed approach is evaluated on three datasets, including CelebA, mammography, and chest X-rays, and demonstrates convincing results.

**Strengths:**

1. The motivation of this work is solid and well-justified. Avoiding spurious correlations in generative models is a challenging task that is worth pursuing.
2. This work provides the necessary prerequisites for understanding the paper in Section 2. The structure of the paper is clear and easy to follow, and the writing is overall clear and coherent.
3. The visualizations of the results are quite convincing.
The evaluation of the generated images in terms of effectiveness and reversibility also makes sense for counterfactual generation tasks.
4. I also appreciate the effort of including medical data, as it is valuable to incorporate datasets that are closer to real-world settings.

**Weaknesses:**

1. The paper lacks prior literature discussing the issue of attribute amplification. I believe a paragraph in sec 2 dedicated to attribute amplification is needed, clarifying how it is defined and summarizing previous studies that have encountered this issue, especially in the context of counterfactual generation.

2. There is also a lack of details on how the CFG model was trained. I am somewhat confused about the training setup: did the authors use only the target attributes for supervision, or were other attribute annotations included as well?

3. There are no numerical results presented in tables; all results are shown in figures, which makes it difficult to assess quantitative values. This could be considered a minor weakness.

4. Regarding reproducibility, it does not appear that the experiments were conducted with multiple random seeds.

**Questions:**

1. I am a bit unsure about why it needs to be a CFG for counterfault generation. As I believe for generating counterfactual, you have a target in mind, for example, change the disease label from 0 to 1, it makes more sense to have a classifier guidance to me. Can you explain the intuition of having CFG here?
Following this question, the biassed results only happen when w is bigger, whether more and more CFM is introduced. If it is purely classifier guided, will there still be such an issue?

---

> ### Author Response · Authors · 2025-11-24
>
> We thank the reviewer for the constructive and thoughtful feedback. We appreciate the positive comments on the motivation, clarity, structure, and evaluations, as well as the recognition of the importance of including real-world medical datasets. Below, we briefly address the main points raised.
>
> ***Lack of prior literature on attribute amplification.***
>
> We agree that Section 2 would benefit from an explicit paragraph describing attribute amplification and summarizing prior work that has observed this phenomenon, and we will add this in a future revision.
>
> ***Training details of the CFG model.***
>
> The full training setup, including the supervision used for all attributes, is already provided in the Appendix. Due to space constraints, we were not able to include the complete procedure in the main text, but we agree that this information should be easier to find. In a future revision, we will make the cross-referencing clearer and add a summary in the main paper while keeping the detailed explanation in the Appendix.
>
> ***Numerical results and tables.***
>
> Numerical results are already included as tables in the Appendix (Tables A.1–A.12), but we agree that relying primarily on figures in the main text may have made these quantitative values less visible. In a future revision, we will make this clearer by surfacing key tables in the main paper or explicitly referencing the Appendix tables to improve readability.
>
> ***Reproducibility and random seeds.***
>
> We will consider adding multi-seed reporting in a future version.
>
> ***Use of CFG for counterfactual generation.***
>
> We use classifier-free guidance because it is the standard and most practical approach in modern conditional diffusion models: it avoids the need to train an external classifier on both clean and noisy samples, which can be expensive, unstable, and dataset-dependent. CFG enables counterfactual generation directly through the model’s semantic conditioning interface, without additional networks.
>
> ***Regarding amplification.***
> The effect we study arises from the score extrapolation in CFG at larger guidance weights. Classifier guidance operates differently, by injecting classifier gradients, and therefore exhibits different failure modes depending on the classifier’s disentanglement and bias. While classifier guidance does not produce the same CFG-style extrapolation, it can still unintentionally affect non-target attributes if the classifier is biased or correlations are present. We will clarify this distinction and will also consider adding a classifier-guided comparison in a future version.

---

### Official Review · Reviewer_4VNU · 2025-10-31

**Soundness:** 3
**Presentation:** 3
**Contribution:** 2
**Rating:** 6
**Confidence:** 4

**Summary:**

The paper proposes Decoupled Classifier-Free Guidance (DCFG), a simple yet effective modification of standard classifier-free guidance (CFG) for counterfactual diffusion models. Standard CFG suffers from the problem of *attribute amplification*, where increasing the guidance strength for one intervention unintentionally alters correlated attributes (e.g., changing “Young” decreases “Male”). DCFG addresses this by splitting attributes into disjoint groups (intervened vs. invariant) and assigning separate guidance weights for each group. This decoupling enables more disentangled, causally faithful counterfactuals without retraining or architectural changes. Experiments on CelebA-HQ, EMBED (mammography), and MIMIC-CXR show that DCFG reduces amplification and improves reversibility compared to standard CFG.

**Strengths:**

- **Practical and elegant idea:** The attribute-split conditioning mechanism is extremely simple and requires minimal changes to existing diffusion pipelines while producing clear improvements.
- **Generality:** The approach is model-agnostic and could extend to other settings.
- **Strong empirical validation:** Results are shown across both natural and medical datasets with quantitative metrics and convincing qualitative examples. The inclusion of reversibility and cross-attribute correlation analysis is helpful and demonstrates the problem concretely.

**Weaknesses:**

* **Clarity of motivation and intuition.**
  The paper tackles an important issue (attribute amplification in classifier-free guidance) but why this happens is not very clearly explained. The reviewer had to re-read [1] and infer the connection independently. Including a brief intuitive explanation, figure or toy example would make the motivation easier to grasp.

* **Simplicity and missing contextualization.**
  The proposed fix of splitting attributes into intervened and invariant groups and applying separate guidance weights is extremely simple and easy to adopt, which is a strength. However, the paper does not discuss whether similar disentangling or conditional-guidance approaches have been explored before, or how this method compares to more sophisticated alternatives for representation disentanglement. A short discussion or empirical comparison would clarify novelty.

* **Missing baselines and ablations.**
  The experiments compare only to standard CFG. It would strengthen the work to include other diffusion-based counterfactual explanation or editing baselines (e.g., [2], [3]).

* **Incomplete evaluation metrics.**
  The paper does not report realism measures such as FID or sFID, nor composition or minimality metrics commonly used in counterfactual image generation. The authors explain the omission of composition in the Appendix, mentioning this in the main text would help. Measuring also minimality through a VLM or a user study, or a brief note on why it is difficult to quantify would make the evaluation more comprehensive.

* **Unaddressed observations.**
  In Figure 1, increasing guidance for *do(Young)* appears to reduce *Male*, likely due to a dataset bias that biases the classifier. This should be discussed and attributed to this bias or another factor. Figure 5 shows similar unexplained behavior (*do(circle)* affecting density AUROC).

* **Minor presentation issues.**
  - Figure 5 would benefit from same interventions between subfigures.
  - Figure 3 is difficult to read and should be split.
  - Equation (5) seems to omit $\tilde{x}$
  - Equation (12) may need a $(1-\omega)$ term for completeness
  - line 284 should reference Appendix C.

[1] Tian Xia, Mélanie Roschewitz, Fabio De Sousa Ribeiro, Charles Jones, and Ben Glocker. Mitigating
attribute amplification in counterfactual image generation. In International Conference on Medical
Image Computing and Computer-Assisted Intervention, pp. 546–556. Springer, 2024.

[2] Guillaume Jeanneret, Loı̈c Simon, and Frédéric Jurie. Diffusion models for counterfactual explana-
tions. In Proceedings of the Asian conference on computer vision, pp. 858–876, 2022.

[3] Preechakul, K., Chatthee, N., Wizadwongsa, S., & Suwajanakorn, S. (2022). Diffusion Autoencoders: Toward a Meaningful and Decodable Representation. IEEE Conference on Computer Vision and Pattern Recognition (CVPR).

**Questions:**

1. Can you provide a clearer intuition, illustrative example or figure for why CFG leads to attribute amplification?
2. Have you tried a version that removes invariant attributes from conditioning entirely?
3. How does DCFG compare with other diffusion-based counterfactual or editing methods such as DiME, or diffusion autoencoders?
4. Can you report or discuss realism metrics (FID/sFID) and briefly justify the exclusion of composition/minimality metrics?
5. Could you clarify the observed cross-attribute effects in Figures 1 and 5 and verify the potential inconsistencies in Equations (5) and (12)?

---

> ### Author Response · Authors · 2025-11-24
>
> We thank the reviewer for the constructive and detailed feedback. We appreciate the positive assessment of the idea, empirical validation, and generality. We agree with the points raised regarding clarity, contextualization, and completeness. Below, we provide brief responses and will incorporate these improvements in a future revision.
>
> ***Clarity of motivation and intuition.***
>
> Thank you for pointing this out. We agree that the intuition behind why classifier-free guidance can induce attribute amplification was not sufficiently clear. In a future revision, we will add a brief intuitive explanation and an illustrative figure to make the motivation more accessible.
>
> ***Comparisons with baselines and ablation studies.***
>
> Approaches such as DiME (classifier-gradient-based–based editing) and diffusion autoencoder methods (latent-space manipulation) operate under substantially different assumptions from our inference-time guidance framework, which is why we initially focused on standard CFG as the most direct baseline. Nevertheless, we acknowledge the value of including these perspectives and additional ablations, and we will incorporate this discussion and more comprehensive comparisons in a future version.
>
> ***Incomplete evaluation metrics.***
>
> Our current experiments focus on quantifying attribute amplification, but we will incorporate these additional measures, i.e. FID, in a future revision. We will also move the explanation regarding composition from the Appendix into the main text for clarity.
>
> ***Unaddressed observations.***
>
> Thank you for highlighting these points. The behavior observed in Figure 1 (do(Young) affecting Male) and Figure 5 (do(circle) influencing density AUROC) reflects both dataset-level attribute correlations and model-side entanglement: strong guidance on one attribute can drift the sample into regions where other attributes are unintentionally affected. We agree this should be discussed more explicitly and will include a brief clarification in a future revision.
>
> ***Removing invariant attributes from conditioning.***
>
> We have tested a variant that removes invariant attributes from the conditioning (i.e., setting their guidance weight to zero) and found that it leads to undesirable behavior. We will clarify this observation in a future revision.
>
> ***Minor presentation issues.***
>
> Thank you for pointing these out. We will correct these presentation issues in a future version.

---

### Author Response · Authors · 2025-11-24

We thank the reviewers and the area chair for their time and thoughtful feedback. We acknowledge the concerns raised and appreciate the opportunity to clarify several points that may have been misunderstood.

We briefly address the key misunderstandings and conceptual clarifications for the record. We have decided to withdraw the paper from further consideration, but we hope this response helps clarify our contribution for any future readers or reviewers.

---

### Note · Authors · 2025-11-24

I have read and agree with the venue's withdrawal policy on behalf of myself and my co-authors.